# Cell-Penetrating Peptide-Mediated Delivery of Gene-Silencing Nucleic Acids to the Invasive Common Reed *Phragmites australis* via Foliar Application

**DOI:** 10.3390/plants14030458

**Published:** 2025-02-05

**Authors:** Qing Ji, Kurt P. Kowalski, Edward M. Golenberg, Seung Ho Chung, Natalie D. Barker, Wesley A. Bickford, Ping Gong

**Affiliations:** 1Bennett Aerospace, Inc., Raleigh, NC 27603, USA; jjessica@iastate.edu; 2U.S. Geological Survey, Great Lakes Science Center, Ann Arbor, MI 48105, USA; kkowalski@usgs.gov (K.P.K.); wbickford@usgs.gov (W.A.B.); 3Department of Biological Sciences, Wayne State University, Detroit, MI 48201, USA; golenberg@wayne.edu; 4Environmental Laboratory, U.S. Army Engineer Research and Development Center, Vicksburg, MS 39180, USA; seung.h.chung@usace.army.mil (S.H.C.); natalie.d.barker@usace.army.mil (N.D.B.)

**Keywords:** RNA interference (RNAi), gene silencing, cell-penetrating peptide (CPP), gene delivery, *Phragmites australis*, phytoene desaturase (*PDS*), γ-zein-CADY, double-stranded RNA (dsRNA), artificial microRNA (amiRNA), antisense oligonucleotide (ASO)

## Abstract

As a popular tool for gene function characterization and gene therapy, RNA interference (RNAi)-based gene silencing has been increasingly explored for potential applications to control invasive species. At least two major hurdles exist when applying this approach to invasive plants: (1) the design and screening of species- and gene-specific biomacromolecules (i.e., gene-silencing agents or GSAs) made of DNA, RNA, or peptides that can suppress the expression of target genes efficiently, and (2) the delivery vehicle needed to penetrate plant cell walls and other physical barriers (e.g., leaf cuticles). In this study, we investigated the cell-penetrating peptide (CPP)-mediated delivery of multiple types of GSAs (e.g., double-stranded RNA (dsRNA), artificial microRNA (amiRNA), and antisense oligonucleotide (ASO)) to knock down a putative phytoene desaturase (*PDS*) gene in the invasive common reed (*Phragmites australis* spp. *australis*). Both microscopic and quantitative gene expression evidence demonstrated the CPP-mediated internalization of GSA cargos and transient suppression of *PDS* expression in both treated and systemic leaves up to 7 days post foliar application. Although various GSA combinations and application rates and frequencies were tested, we observed limitations, including low gene-silencing efficiency and a lack of physiological trait alteration, likely owing to low CPP payload capacity and the incomplete characterization of the PDS-coding genes (e.g., the recent discovery of two *PDS* paralogs) in *P. australis*. Our work lays a foundation to support further research toward the development of convenient, cost-effective, field-deployable, and environmentally benign gene-silencing technologies for invasive *P. australis* management.

## 1. Background

*Phragmites australis* (Cav.) Trin. ex Steud., also known as the common reed, is one of the most widely distributed perennial grasses [1] (also refer to https://www.gbif.org/species/5290149 (accessed on 21 January 2025)). It is tolerant of acidic, saline, and alkaline environments; grows on level ground in both tidal and non-tidal marshes, lakes, swales, and the backwater areas of rivers and streams; and can be found growing on most soil textures, from fine clay to sandy loam [2]. The subspecies *P. australis* ssp. *australis* (*P. australis*) is not native to North America. It may grow up to 6 m tall and can spread horizontally 5 m or more per year by clonal growth via stolons and rhizomes, thereby forming dense stands [1,3]. Its vigorous vegetative growth and large seed production enable the grass to outcompete native vegetation, which, in turn, greatly reduces plant and animal biodiversity, decreases property values, and impairs the recreational use of wetlands and shorelines [4]. In addition, there is evidence that *P. australis* secretes gallic acid (GA), which is then photo-degraded into mesoxalic acid (MOA) [5]. Both GA and MOA act as phytotoxins to ward off encroachment by neighboring plants and thus may provide *P. australis* with a competitive edge in its habitats. However, results from a later study did not support the role of GA in the invasion success of nonindigenous *P. australis* [6]. The invasiveness of non-native *P. australis* has been well documented (e.g., its rapid invasion into North America [7,8] and especially the Great Lakes region [9,10]), and it has been listed by several states in the U.S. as a noxious weed (refer to https://plants.usda.gov/plant-profile/PHAU7 (accessed on 21 January 2025)).

Once established, effective control of the invasive *P. australis* is extremely hard using a single technique due to its rapid growth and robust and regenerative root system [11]. Currently, *P. australis* control relies on herbicide (glyphosate) treatment, hydrological flooding, mechanical removal, and prescribed burning or deep “root burns” [12,13] (also refer to https://www.greatlakesphragmites.net/management/ (accessed on 21 January 2025)). These methods require repeated treatments for several seasons or years to yield a desirable outcome, which could lead to environmental pollution and other problems. Novel cost-effective and environmentally benign methods could support the management of this invasive species. Here, we focus on ribonucleic acid interference (RNAi)-based gene-silencing approaches as a potential means of novel genetic biocontrol of *P. australis*.

Gene silencing (also known as gene knockdown) is a general term describing the interruption or suppression of gene expression at transcriptional or post-transcriptional/translational levels. A wide variety of strategies have been applied to repress gene expression, including hybridizing to target messenger RNA (mRNA), catalyzing the cleavage of target mRNA, and binding to target proteins (refer to https://www.ncbi.nlm.nih.gov/probe/docs/applsilencing/ (accessed on 21 January 2025)). In plants, four distinct RNAi pathways have been discovered, including post-transcriptional microRNA (miRNA), trans-acting small interfering RNA (tasiRNA), RNA-directed deoxyribonucleic acid (DNA) methylation, and exogenic RNA silencing [14,15]. These mechanisms have in turn been exploited for developing artificial RNA-silencing technologies, such as hairpin RNA (hpRNA), double-stranded RNA (dsRNA), artificial microRNA (amiRNA), intrinsic direct repeat, 3′-untranslated region (UTR) inverted repeat, artificial tasiRNA, and virus-induced gene-silencing (VIGS) technologies [15]. Gene silencing has been widely used to (1) generate or enhance disease resistance to various pathogens like viruses, fungi, and insect pests [16,17,18]; (2) improve nutrition value in food crops [19,20,21]; (3) promote desirable agronomic traits [22,23]; and (4) increase the shelf life of tomatoes [24], kiwis [25], and strawberries [26]. Recently, gene silencing has been investigated as a novel approach for the control of pests and pathogens in a sequence-specific fashion without genetically modifying the host plant [27,28,29]. For instance, the injection of dsRNA targeting the inhibitor of apoptosis into the invasive starry sky beetle (*Anoplophora glabripennis*), native to Asia, caused 90% death of the larvae and 100% death of the adults [30]; FANA ASOs (2′-deoxy-2′-fluoro-arabinonucleic acid antisense oligonucleotides) effectively suppressed *Candidatus* Liberibacter asiaticus or CLas (a bacterium causing citrus greening) transmission and reduced its pathogenic impact on citrus trees [31]; and sugarcane mosaic virus-based VIGS constructs efficiently silenced endogenous genes putatively coding for jasmonate-isoleucine (JA-Ile) hydroxylase and JA-Ile hydrolase in maize, leading to increased herbivore resistance [32].

A bottleneck factor in the application of gene-silencing technologies in plants is the delivery of gene-silencing agents (GSAs) into plant cells. Gene silencing has been studied mostly by transforming plants with vectors producing GSAs, such as dsRNA, hpRNA, and small interfering RNA (siRNA) via *Agrobacterium* [33,34,35,36] or bombardment [37]. These methods require plant transformation, which means longer turn-around times and limited success in species recalcitrant to transformation. Virus-induced gene silencing works by infecting plants with modified viruses carrying a transgene cargo that produce siRNAs, which do not require plant transformation [38]. However, VIGS is host-specific, and the number of viruses that can be engineered successfully for this purpose is limited so far. Other delivery methods have been explored to overcome these problems, including nanosecond pulsed laser-induced stress waves [39] and nanomaterial-based delivery vehicles [40,41], such as layered double hydroxide (LDH) clay nanosheets [42], single- or multi-walled carbon nanotubes (SWCNT or MWCNT) [43,44,45], carbon nanodots (CND) [46], and other nanostructures like tetrahedron, hairpin-tile monomer, and nanostring [47]. Nevertheless, we have not seen documented applications of the nanoparticle-mediated delivery platforms in plant species other than a few model species such as thale cress (*Arabidopsis thaliana*) and benthi (*Nicotiana benthamiana*).

The cell-penetrating peptide (CPP)-based gene delivery system is an emerging technology that has been widely used in the intracellular delivery of molecules, varying from small molecular drugs to macromolecular DNA, RNA, and peptides [48], and for many different cell types including plant cells [49]. As a distinct group of peptides, CPPs are often composed of 5 to 30 amino acids [50], facilitating the transport of cargos across various cellular boundaries via multiple mechanisms, including endocytosis, direct penetration, and translocation [48]. In addition, CPPs can be designed to target the cytoplasm, the nucleus, and specific cellular organelles such as the endoplasmic reticulum, Golgi apparatus, mitochondria, and chloroplasts [51]. After efficient cellular internalization, CPPs can release their cargo into the cytosol to promote the desired biological effect [48]. Numerous natural and synthetic peptide sequences possessing cell-penetrating properties have been documented since the first reports of two landmark CPPs, Penetratin and Tat, over two decades ago [52,53]. Based on their sequence, polarity, and hydrophobicity, CPPs can be grouped into five classes, cationic, amphipathic, hydrophobic, proline-rich/antimicrobial, and bipartite/chimeric [51,54], with cationic CPPs being the most studied class, representing more than 75% of the published work [51]. Cell-penetrating peptides can be used to translocate plasmid DNA to induce gene expression in plants [55]. They can also deliver siRNA and microRNA molecules for post-transcriptional gene silencing using CPP–nucleic acid conjugates or CPP–nucleic acid carriers [56,57]. Of the hundreds of CPPs known to date, only a few CPPs have demonstrated the ability to transport cargo into plant cells (Table 1). These CPPs have been reported to penetrate and transport macromolecules such as lysozyme, proteins, DNA, plasmids, and other nucleic acids into protoplasts or intact plant cells. Target plants include tomato, corn [58,59], onion [58,59,60], mung bean [61], soybean [61], triticale [62,63], tobacco [55,64,65,66], rapeseed [67], wheat [67], thale cress [49,55,56,68], and carrots [49]. The advantages of using CPPs as the carrier for gene delivery include the ease of synthesis and use, low or no cytotoxicity, versatility in cargo composition and intracellular target sites, no interference with cargo functionality, and biodegradability [48,69]. However, nearly all the studies reported so far have been conducted in vitro at the cellular level. Very few in vivo studies have been conducted to date to test CPP-mediated gene silencing in model plant species (e.g., tomato [57] and thale cress [56]). In this study, we attempted to develop a CPP-based gene-silencing system as a novel biotechnology for the control of invasive *P. australis* plants.

## 2. Methods

### 2.1. P. australis Growth Conditions

Invasive *P. australis* plants were propagated from rhizomes in the U.S. Army Engineer Research and Development Center (ERDC) greenhouse located in Vicksburg, Mississippi. The rhizomes were derived from field specimens collected at the Cedar Point National Wildlife Refuge in Oregon, OH, and the Ottawa National Wildlife Refuge in Oak Harbor, OH, both locations on the western basin of Lake Erie. The rhizome was cut into smaller segments, each with >3 internodes and at least one active meristem. The pre-cut rhizomes were planted in commercial soil (Miracle-Gro^®^ potting mix) in 1 L pots, and the pots were placed in a tray without drain holes. Standing water was achieved by adding 3–4 cm of water to the tray every two days. Each pot of plants was initially fertilized with a ½ tsp of Osmocote^®^ Smart-Release^®^ plant food (NPK 14-14-14) and then with a ¼ cup of nutrient solution created with Peter’s fertilizer (NPK 20-20-20) and 10% chelated iron (¼ cup each dissolved in 20 gallons of water) every two weeks. The plants grew in the greenhouse under a 15 h/9 h (light/dark) cycle at 25 °C.

### 2.2. Cell-Penetrating Peptide (CPP) Design and Synthesis

The schematic structure of our newly designed fusion CPPs was CPP1–spacer– (CPP2–spacer)–endosomal escape domain (EED) [76,77]. A total of five new CPPs were designed in house and chemically synthesized by ABI Scientific Inc. (Sterling, VA, USA). The five CPP sequences are shown in Table 2. Each CPP was synthesized with or without a fluorescein isothiocyanate (FITC) tag at the N-terminal. Two of the five CPPs, γ-zein-CADY and R9-CADY, were fusion peptides consisting of two CPPs connected by a spacer amino acid glycine (G). Each CPP contained a GFWFG tag at the C-terminus, which functioned as an EED to avoid being trapped in endosomes and degraded by lysosomal enzymes, leading to enhanced intracellular delivery [78].

### 2.3. P. australis Phytoene Desaturase (PaPDS)-Expressing Plasmid Construction and dsRNA Production

A 277-bp insert coding for a protein-coding segment of a 788-bp putative *P. australis* phytoene desaturase *(PaPDS*) transcript was cloned into a Promega pGEM^®^-T Easy Vector to make the plasmid pGEM-PaPDS at the multiple cloning region flanked by recognition sites for a wide variety of restriction enzymes (e.g., *SphI*, *EcoRI*, *SalI*, and *NotI*) and by a T7 and a Sp6 promoter on the up- and downstream, respectively. Appendix A describe in detail the production protocol of dsRNA transcripts targeting the putative *P. australis PDS* gene (i.e., dsRNA*_PaPDS_*). Appendix A provides the *PaPDS* sequence and the pGEM®-T Easy Vector map. Briefly, the *P. australis PDS* expression construct was first transformed into competent *Escherichia coli* (*E. coli*) cells (JM109 from Promega, Madison, WI, USA). Positive clones of transformed cells streaked on LB/Amp plates were picked and grown in LB/Amp medium overnight. The cells were then harvested, and plasmids were isolated. Plasmid linearization was achieved by restriction enzyme digestion, i.e., *Sal*I digestion for linear transcripts with the T7 promoter or *Sph*I digestion for linear transcripts with the SP6 promoter. In vitro transcription was conducted using an Ambion^TM^ MAXIscript^TM^ SP6/T7 Transcription Kit (catalogue# AM1322, Thermo Fisher Scientific, Waltham, MA, USA). The resulting two single stranded RNAs were annealed together by heating to 70 °C for 10 min and cooling down slowly to room temperature. The generated dsRNA was further purified using a QIAGEN RNeasy mini kit (catalogue# 74104, Germantown, MD, USA). The concentration of final dsRNA was quantified by a NanoDrop One Spectrophotometer (NanoDrop Technologies, Wilmington, DE, USA). The Silencer™ siRNA Labeling Kit with FAM dye (Thermo Fisher Scientific catalogue# AM1634; emission wavelength of 518 nm) was used to label the dsRNA for CPP-mediated gene delivery testing and visualization.

### 2.4. Design and Synthesis of Antisense Oligo (ASO) and Artificial microRNA (amiRNA)

For the antisense oligo (ASO) design, the putative *PaPDS* sequence was first entered into the online InvivoGen siRNA Wizard Software 3.1 (http://www.invivogen.com/sirnawizard/design.php (accessed on 21 September 2020)) to design siRNAs. The reverse complementary DNA sequences of the siRNAs were selected as ASO candidates and synthesized by Integrated DNA Technologies Inc. (IDT, Coralville, IA, USA). During ASO synthesis, the last three bases on both the 3′- and 5′-ends were modified by substituting a sulfur atom for a non-bridging oxygen atom in the phosphate backbone of an oligonucleotide in order to enhance ASO stability because the phosphorothioate modification could render the internucleotide linkage resistant to nuclease degradation (https://www.idtdna.com/pages/education/decoded/article/modification-highlight-modifications-that-block-nuclease-degradation (accessed on 21 January 2025)). The amiRNA sequences were designed to silence *PaPDS* using the Web MicroRNA Designer 3 (WMD3) online tool (http://wmd3.weigelworld.org/cgi-bin/webapp.cgi (accessed on 21 January 2025)), and the top five amiRNA candidates and a fluorescently labeled Atto550-amiDNA (instead of amiRNA for cost-saving and stability considerations) were ordered from IDT. All the sequences of the 12 phosphorothioated ASOs and 6 amiRNA/DNA are listed in Table 3.

### 2.5. Cell-Penetrating Peptide/Gene-Silencing Agent (CPP:GSA) Complexation and Topical Application

Cell-penetrating peptides were dissolved in 50% dimethyl sulfoxide (DMSO) solution and then diluted in deionized water to make the final concentration of 1 µg/µL. Gene-silencing agents were dissolved in nuclease-free water at 1 µmol/µL. The CPP and GSA were mixed at proper ratios by gently tapping on the tube and incubated on ice for at least 1 h. Immediately before application, an equal amount of infiltration medium composed of 10% sucrose (*w*/*v*) and 0.08% Silwet^®^ L-77, a non-ionic surfactant, was added to the conjugation mixture. Young *P. australis* plants with 6–8 leaves were selected, with one leaf per plant receiving the treatment. A volume of 30 µL of treatment solution was loaded on the adaxial (upper) side of a leaf and allowed to slowly spread across the whole leaf surface without dropping off. Each treated leaf was marked by affixing a sticky label to the stem close to the leaf. To determine the ability of the chosen CPP to carry GSA into plant cells, FAM-labeled dsRNA*_PaPDS_* in complex with non-labeled γ-zein-CADY or Atto550-labeled amiDNA*_PDS_*_-1_ in complex with FITC-labeled γ-zein-CADY was applied in vivo to the *P. australis* leaf surface, and the treated leaves were observed microscopically.

### 2.6. Preparation and Observation of Leaf Cross-Section Slides

After treatment (varying from 1 to 8 days depending on the experiments), fresh *P. australis* leaf samples were harvested, washed with 10% phosphate buffer solution, rinsed with Millipore-purified water, pat dried, and kept on ice before further processing. A small section (approximately 0.3 cm in width) was cut off across the midrib of the fresh leaf and embedded in a tissue freezing medium (Leica Biosystem, Buffalo Grove, IL, USA; Catalogue# 14020108926). Using a Leica CM1860 Cryostat (Deer Park, IL, USA), the embedded sample was cryosectioned to obtain cross sections of 20 µm in thickness. After discarding the first five cross sections, the remaining ones were gently transferred onto a cold Superfrost™ Plus Microscope Slide (Thermo Fisher Scientific, catalogue# 4951PLUS4) and covered with a cold coverslip (Thermo Fisher Scientific, catalogue# 22X40-1). The prepared slides were stored in the dark at −20 °C until visualization using a regular fluorescence microscope (i.e., Leica DM IL LED Inverted Fluorescence Microscope (Deerfield, IL, USA)) or a confocal fluorescence microscope (refer to the next section for more details). At least three cross-section slides were prepared from different sections of a sampled leaf, and at least three leaves were examined per treatment group.

### 2.7. Confocal Microscopy Observation

We observed autofluorescence emissions in the range of 500–550 nm when checking the FITC signals in untreated *P. australis* leaves, especially in those leaves with either physical damage or insect bites, which was consistent with previous observations that the fluorescence might stem from the secondary metabolites caused by stress [79,80]. To solve this problem, we enhanced greenhouse management to eliminate insects, used filtered water for irrigation, and increased the fertilizer application to avoid malnutrition in the testing plant materials. More importantly, we replaced the regular fluorescence microscope carrying a broad pass filter with a confocal microscope that had a very narrow filter range to filter out the background noise of fluorescence signals. As a result, the fluorescence signals from the untreated control tissue samples using the FITC channel filter were minimal. Briefly, we used a Zeiss LSM 510 META confocal microscope (Jena, Germany) at the Mississippi IdeA Network of Biomedical Research Excellence (INBRE) Imaging Facility, housed in the University of Southern Mississippi, Hattiesburg campus, to observe the fluorescence signals of fluorescein isothiocyanate (FITC)-labeled CPP (green) and Atto550-labeled GSA (red). The imaging system combined an expanded laser system with an advanced META detector, which allowed for multi-fluorescence imaging and the separation of fluorochromes unable to be imaged together on other instruments. The META detector also allowed greater separation of background autofluorescence from fluorescent dye emissions. Z-stack images were also taken from the specimen at 1–2 μm intervals, and a 3D rendering was generated for a greater depth of field because the image area along the x- and y-axes remained the same, but the distance from the objective (*z*-axis) was different for each image.

### 2.8. RNA Extraction, Reverse Transcription, and Quantitative Polymerase Chain Reaction (qPCR)

Fresh *P. australis* leaf samples were harvested, washed, and dried, followed by snap freezing in liquid nitrogen and subsequent storage at −80 °C. The snap frozen leaf tissue was ground into a fine powder with a mortar and a pestle in liquid nitrogen. Then, RNA was extracted using a QIAGEN RNeasy Plant Mini Kit (catalogue# 74904), and the total RNA concentrations were measured using the NanoDrop One Spectrophotometer (NanoDrop Technologies). The integrity of total RNA for each sample was assessed using an Agilent 2100 Bioanalyzer (Palo Alto, CA, USA). The total RNA (2.5 µg/sample) was reverse transcribed into complementary DNA (cDNA) using a SuperScript™ IV VILO™ Master Mix with ezDNase™ Enzyme (Thermo Fisher Scientific, catalogue# 11766050). The quantitative polymerase chain reaction (qPCR) assay was conducted using SYBR green PCR master mix (Thermo Fisher Scientific, catalogue# 4309155) with 18 µL of 1:25 diluted cDNA input and 0.5 µM qPCR primers in a 20 µL reaction. The reaction mixture was added to an optical 384-well plate (Thermo Fisher Scientific, catalogue# 4483285) and analyzed using a QuantStudio™ 7 Flex Real-time PCR system (Thermo Fisher Scientific) set to the following thermal program: 10 min at 95 °C, 40 cycles of 95 °C for 15 s, and 60 °C for 1 min. Primer specificity was verified by melting curve analysis (95 °C for 15 s, 60 °C for 1 min, and 95 °C for 15 s). The qPCR data analysis was conducted using the instrument’s pre-installed and licensed QuantStudio™ Real-Time PCR Software v1.7.1. The qPCR primer sequence information is presented in Table 4.

### 2.9. Data Analysis

The qPCR results were analyzed using LinRegPCR (version 2018.0) [81], and the relative expression was calculated by normalizing the target *PDS* gene expression level to that of the reference gene (actin). Statistical analyses for assessing the effects of the experimental treatments on *PDS* expression were performed using the R package installed in Windows (https://cran.r-project.org/doc/manuals/r-patched/R-admin.html (accessed on 21 January 2025)). They included *t*-tests and ANOVA tests, with the goal of identifying significant differences (*p* < 0.05) between treatment groups and controls. The Tukey *post hoc* test was performed to identify further which specific group differed significantly from the controls (*p* < 0.05) when significance was detected in the ANOVA test. The raw data are available at https://doi.org/10.5066/P13ETPDI.

## 3. Results

The following results present microscopic evidence of CPP penetration into and CPP-mediated uptake of GSAs by *P. australis* leaves. The gene expression results are provided to show the inhibitory effects of three different classes of GSAs (i.e., dsRNA, amiRNA, and ASO) on the target gene *PDS* in the treated leaves.

### 3.1. Tissue Uptake of CPPs via Topical Application on P. australis Leaf

All five FITC-labeled CPPs (R9, γ- zein, CADY, R9-CADY, and γ- zein-CADY) demonstrated the capability of penetration into *P. australis* leaf tissue when they were applied ex vivo individually at 5 nmol of CPP/leaf to the adaxial (upper) surface of freshly cut leaves along with the infiltration medium (Figure 1). The application rate of 5 nmol of CPP/leaf was determined in preliminary experiments and implemented throughout the present study. After one hour of foliar application, FITC (green) fluorescence was observed in all leaf cross-section samples except the infiltration medium-only control (no CPP). The same phenomenon was observed for *P. australis* leaves treated in vivo with FITC-labeled CADY and γ- zein-CADY (results not shown in graphs). However, the five CPPs displayed differential potential, with R9 being the weakest penetrator, as most of the fluorescence remained on the leaf surface (Figure 1). Comparatively, R9-CADY, CADY, and γ-zein-CADY had more fluorescence signals inside the tissue relative to the other two CPPs. Based on these results, γ- zein-CADY was selected for further experiments in this study.

### 3.2. CPP-Mediated In Vivo Uptake of GSA by P. australis Leaf

The CPP-mediated GSA delivery is mainly determined by two factors: (1) the net positive charge carried by a CPP and the net negative charges carried by a GSA, and (2) the loading capacity set by the actual CPP:GSA charge ratio. Although the net charges of CPPs and GSAs are unchangeable, the loading capacity is adjustable. Here, we varied the CPP:GSA charge ratio from 1:1 to 50:1 to determine the appropriate loading capacity. For the CPP:dsRNA (at a charge ratio of 1:1) treated leaves, the delivery efficiency was quantified by counting green, fluorescent dots (FAM-labeled dsRNA) in an undamaged cross section per slide and three slides per leaf. As shown in Figure 2, the number of internalized fluorescent dots in the treated leaves was significantly higher (*p* < 0.05 or 0.01) than that in the control leaves (background autofluorescence) and increased as the treatment duration was prolonged from 1 day to 3 or 5 days. For the CPP:amiDNA treatments (at the charge ratio of 50:1 or 6.25:1), amiDNA was labeled with Atto550, whereas γ- zein-CADY was either labeled with FITC or not labeled, and the cross-section slides were made for 1 DPT (day post treatment), 4 DPT, and 7 DPT leaf samples for confocal microscopy observation. Analysis of the collected confocal microscopic images showed that the CPP-mediated delivery efficiency of amiDNA was the highest at 4 DPT, which is consistent with the CPP:dsRNA results (Figure 2). We present a collection of the 4 DPT images in Figure 3. The treatment solution of the Atto550-amiDNA*_PDS-1_*:FITC-γ-zein-CADY complex was used as a positive control (Figure 3a–c). The Atto550-amiDNA*_PDS-1_*-only treatment served as the negative control (Figure 3l–o), which showed no signal in either the green (Figure 3m) or the red (Figure 3n) channel. The FITC-labeled CPPs were observed as green signals (Figure 3a,e,i,m), and Atto550-labeled amiDNA*_PDS_*_-1_ was observed as red signals (Figure 3b,f,j,n), where overlap in the fluorescence among the labels produced a yellow signal indicating the co-location of γ-zein-CADY and amiDNA*_PDS_*_-1_ (Figure 3c) and complex formation among the molecules. Complex formation was a key pre-requisite observation for moving forward with the CPP–GSA gene-silencing assays. In the treated leaf samples, the overlap of the red and green signals was again detected, although the co-location of γ- zein-CADY and amiDNA*_PDS_*_-1_ was sparse (Figure 3g). Specifically, the CPP was well distributed in the leaf tissue, whereas the GSA was minimally present, likely due to the low GSA:CPP charge ratio (1:50) and the limited amount of GSA:CPP complex solution that could be applied per leaf, but the regions where the GSA was found were co-located with the strongest presence of CPP. When the GSA:CPP ratio was increased to 1:6.25, significantly more amiDNA was internalized (Figure 3j,k). This result indicated that the complex of γ-zein-CADY and amiDNA*_PDS_*_-1_ could penetrate the plant cells. However, the optimal ratio between CPP and GSA was unknown, and therefore, in the following assays, we adjusted the ratio with the goal of improving GSA penetration efficiency (refer to the next section).

### 3.3. Determination of Appropriate CPP:GSA Ratios

To determine the optimal or appropriate CPP:GSA charge ratio, a fixed volume of non-labeled γ-zein-CADY was complexed with varying volumes of Atto550-labeled amiDNA*_PDS_*_-1_ to achieve the target CPP:GSA charge ratio of 1:1, 2:1, 4:1, 6.25:1, 12.5:1, and 25:1. Plant-tissue penetration was investigated in leaves sampled at 4 DPT using confocal microscopy, where results indicated increasing amiDNA*_PDS_*_-1_ penetration with a decreased CPP:GSA (i.e., γ-zein-CADY:amiDNA*_PDS_*_-1_) ratio (Figure 4). The z-stack images also indicated that the amiDNA*_PDS_*_-1_ signal appeared inside the leaf tissue, instead of being localized on top of or below the leaf surface (Figure 5). The most abundant amiDNA*_PDS_*_-1_ signal was observed in the 1:1 (CPP:GSA) treatment; however, a significant amount of the fluorescence signal was found outside the leaf tissue at that ratio, indicating a potential saturating concentration of amiDNA as the cargo of the free γ-zein-CADY (Figure 4g). When the CPP:GSA ratio was above 12.5:1, the amount of internalized amiDNA*_PDS-1_* was nearly negligible (Figure 3f and Figure 4b), suggesting that the CPP:GSA complex concentration was too low. Therefore, one should be able to control the desired amount of internalized GSA by adjusting the CPP:GSA ratio between 12.5:1 and 2:1 (Figure 4c–f) in a semi-quantitative fashion.

### 3.4. Screening-Effective GSAs for Gene Silencing in P. australis

The gene-silencing potential of dsRNA*_PDS_* was tested in two experiments. The dsRNA*_PDS_* was loaded with γ-zein-CADY at a ratio of 1:1 (CPP:GSA) on young *P. australis* leaves. In the first experiment, a statistically significant 38% reduction in transcriptional gene expression (*p* < 0.05) was observed in the treated leaf samples at 5 DPT (Figure 6a). However, this result was not replicable in the second experiment even though it was extended to 8 DPT (Figure 6b). This discrepancy might be due to the FAM-labeling in the first experiment, which protected the labeled dsRNA from nuclease degradation.

Next, we designed a total of 12 ASOs and 5 amiRNAs targeting *PDS* (refer to Table 3 for sequences) and had them synthesized by a commercial vendor. Three pools of ASOs were created with equal amounts of total ASOs in each mixture. Pool A was made up of ASO-1 through ASO-12; pool B included ASO-1 through ASO-6; and pool C contained ASO-7 through ASO-12. We created pools B and C to reduce the dilution factor (1/12 in pool A vs. 1/6 in pools B and C) for each constituent ASO and narrow down the effective ASO candidates (from 12 to 6). The three ASO pools with the same total ASO amount (1 nmol/treatment) were used to treat the *P. australis* plants in combination with γ-zein-CADY for 4 or 7 days. The RT-qPCR results indicated that on day 4, the relative expression of the *PDS* gene in the pool C treatment, but not in the pool A and pool B treatments, was significantly lower than in the γ-zein-CADY-only vehicle control (25% inhibition, *p* < 0.01) (Figure 7a). To narrow down the exact ASO causing the reduced gene expression, pool C was further split into two subgroups to treat the *P. australis*, one with ASO-7, -8, and -9 and the other with ASO-10, -11, and -12. However, neither a statistically significant reduction in target *PDS* gene expression (*p* > 0.05) nor observable phenotypic symptoms were detected in these pooled treatments. The inconsistent results might be due to inefficient or unstable binding between the CPP and GSA as well as the enzymatic degradation of ASOs (Figure 7b).

Similar to the testing with ASOs, a pool containing all five amiRNAs was used in combination with γ-zein-CADY to treat the *P. australis*. Again, no statistically significant reduction in *PDS* gene expression was observed in the pooled amiRNA treatment at 2, 4, and 7 DPT (*p* > 0.05) (Figure 8). We also conducted two single amiRNA (amiRNA*_PDS_*_-1_ and amiRNA*_PDS_*_-2_) treatments, where similar results were obtained with no significant decrease in gene expression (*p* > 0.05) or phenotypic symptoms in any of the treatments (Figure 8). We picked amiRNA*_PDS_*_-1_ and amiRNA*_PDS_*_-2_ for single amiRNA treatments because they ranked the highest among the top five amiRNAs selected for testing.

Furthermore, we tested the gene-silencing effects of repeated foliar applications of GSAs, including daily treatment for 4 consecutive days with CPP (γ-zein-CADY) alone, CPP + ASO pool C (ASO-7 to ASO-12), CPP + amiRNA pool (amiRNA*_PDS_*-1 to amiRNA*_PDS_*-5), or the alternation of CPP + ASO pool C and CPP + amiRNA pool. Among the four treatments, the alternated application of amiRNA and ASO showed a significant decrease (20%) in relative *PDS* gene expression in comparison with the CPP-only control (*p* < 0.05), while the other two treatments did not have any statistically significant effect on *PDS* gene expression (*p* > 0.05) (Figure 9).

We made two CaMV 35S promoter-controlled amiRNA*_PDS_* expression plasmids by replacing the ccdB cassette on the pTAC plasmid [82] with amiRNA*_PDS-1_* or amiRNA*_PDS-3_*-expressing cassettes (Appendix A). The amiRNA*_PDS-1_* expressing plasmid was delivered using CADY or γ-zein-CADY at a 1:1 or 1:50 (GSA:CPP) charge ratio, respectively, to treat the *P. australis* leaves. When CADY was used as the delivery vehicle, no significant gene-silencing effect was detected on either day 1 or day 3 post treatment (*p* > 0.05) (Appendix A). When γ-zein-CADY was applied at a much higher charge ratio, the CPP–plasmid conjugate did not affect *PDS* expression if compared with the CPP-only control even though it significantly enhanced *PDS* expression if compared with the plasmid-only control at DPT 1 (*p* < 0.01). However, this stimulatory effect disappeared at DPT 4 and DPT 7 (Appendix A). Considering the large size of the amiRNA*_PDS_*-expressing plasmids (>10 kb), which may be an obstacle for CPP-mediated delivery, we PCR-amplified the amiRNA*_PDS_*-expressing cassette from the 35S promoter to the NOS terminator that flanked the amiRNA insert sequence (Appendix A) and directly loaded it with γ-zein-CADY. Again, no statistically significant gene silencing was detected at the two sampling time points (*p* > 0.05), i.e., DPT 4 and DPT 7 (Appendix A).

## 4. Discussion

In this study, we determined that the FITC-labeled CPP, γ-zein-CADY, can carry Atto550-labeled DNA oligos into *P. australis* leaf tissue in vivo. We also demonstrated that gene silencing can be achieved by delivering ASOs or a combination of ASOs and amiRNAs into *P. australis* leaf tissue via γ-zein-CADY. The effectiveness of CPPs in delivering gene-silencing agents has been widely reported; however, most of the studies have been conducted in in vitro cellular assays [83,84,85]. To date, very few reports can be found demonstrating the delivery of gene-silencing agents or transgene-expressing cassettes into intact tissues, especially in live plants [56,68]. To the best of our knowledge, this study is the first in vivo CPP-mediated gene-silencing investigation in a monocot plant. Monocots are known for their recalcitrance to in situ genetic transformation due to multiple reasons, such as cell wall chemistry, wound response, callus formation, and regeneration [86]. The method described herein can deliver nucleic acids into host cells without going through more complicated gene transformation processes and thus break the limitation of traditional gene transformation.

Naked GSAs have been shown to induce gene-silencing effects in pathogens and pests when applied to plants [28]. All the experiments included the naked GSA control (GSA + IM) group in addition to other controls (e.g., blank/mock control, IM only, and CPP + IM only; refer to Figure 7b and Appendix A for examples). We chose not to present the naked GSA control (as well as the blank control and IM-only control) in some of the figures because only the CPP + IM control allowed us to make statistical comparisons to derive the effects of CPP-delivered GSA (i.e., GSA + CPP + IM) treatment; so, we always presented the CPP + IM control. Our results showed that the naked GSA control did not differ significantly from the other control groups (*p* > 0.05), suggesting that naked GSAs could not silence their target gene *PDS*. This is consistent with other studies (e.g., [46]) and was supported by evidence that naked GSAs could not internalize plant cells (Figure 3l–o), which is required to exert any gene-silencing effects.

The *PDS* gene encoding the phytoene desaturase enzyme is involved in the carotenoid biosynthesis pathway [87]. VIGS-induced *PDS* silencing has been reported in tobacco, pepper, and tomato, and the silenced plants usually exhibited photo bleaching in the leaves and/or fruits [88,89,90]. Due to its distinct phenotype, *PDS* is usually chosen as a biomarker gene for gene silencing. However, no *PDS* silencing in *P. australis* has been reported until the present study. Here we designed three classes of GSAs (i.e., ASO, dsRNA, and amiRNA) to knock down the *P. australis PDS* gene based on the limited sequence information available at the time of this study, spanning from 2018 to 2021, in an effort to induce the photobleaching symptom in the treated *P. australis* plants. In this study, no visible photobleaching was observed in any of the treated leaves, despite statistically significant lower *PDS* transcription (i.e., 20%~38% suppression) being detected as a result of GSA treatment in multiple experiments. The lack of the expected phenotypic response may be attributed to the existence of two paralogs or alternative splice variants, *PaPDS3_1* (Gene ID: PauIOH1f_m1s_s0056_00390.1t; CDS 1626 bp) and *PaPDS3_2* (Gene ID: PauIOH1f_m1s_s0012_05080.1m; CDS 2289 bp), in the recently published invasive *P. australis* subspecies genome [91], which was deposited in and is retrievable from the CoGe database at https://genomevolution.org/coge/ (accessed on 21 January 2025). These two assembled cDNA models are much longer than the 788-bp putative partial *PDS* transcript and may be translated into 542-aa and 763-aa proteins, respectively. Based on the homology of the conserved *PDS* gene, its ortholog, *PDS3*, in *Arabidopsis thaliana* has three splice variants—AT4G14210.1, AT4G14210.2, and AT4G14210.3—derived from a 5.33-kb genomic region and coding for a 566-aa protein (refer to https://www.arabidopsis.org/servlets/TairObject?name=AT4G14210&type=locus (accessed on 21 January 2025)), whereas another *PDS* ortholog in Foxtail millet (*Seraria italica*) is also 5.37 kb in length and codes for a 571-aa protein (Appendix A). These pieces of genetic and genomic information indicate that more work is warranted to further characterize the PDS-coding gene in *P. australis* to improve the design of GSAs to repress *PDS* gene expression and produce the desired photobleaching phenotype.

Out of three pools of ASOs, we identified a group of six unique ASOs (ASO-7 through ASO-12) that can silence *PDS* gene expression in *P. australis*. However, no gene silencing can be found when further dividing this group into two subgroups of ASOs. This observation may indicate that the specific mixture of ASOs acted synergistically to silence the *PDS* gene. Furthermore, we analyzed the location of those ASOs in the *PDS* gene sequence and found that four out of six ASOs (ASO-9 to ASO-12) in this group were located at the 3′ untranslated region (UTR), with the remaining two (ASO-7 and ASO-8) in the protein-coding region (Appendix A). There are two major mechanisms for ASO-induced gene silencing: (1) hybridizing with a target RNA:Rnase H-dependent mechanism and (2) a steric hinderance mechanism [92]. The former works by degrading mRNA, while the latter operates by preventing translation [92]. An additional mechanism relies on the inhibition of 5′ cap formation and alternative RNA splicing [93]. It has been reported that the translation initiation region (TIR) is where most of the identified ASOs target genomic sequences [94,95], whereas the coding region and 3′UTR sequences have also been identified as targets [96,97]. Due to incomplete sequence information for *P. australis* available at the time of this work, we did not design any ASOs targeting TIR for the *P. australis PDS* gene. Instead, we designed 12 ASOs targeting the coding region as well as the 3′ UTR. Our results indicated that a better silencing effect may be achieved by applying ASOs targeting both the coding region and the 3′UTR, where both mRNA degradation and translation blocking likely function to repress target *PDS* gene expression synergistically.

The most effective *PDS* gene-silencing result was achieved at DPT 4 and decreased by DPT 7, indicating that the decrease in transcriptional expression is transient over time. This observation may result from a limited amount of GSA being “dosed” into cells, which are likely to have a limited half-life based on the stability of the specific GSA molecular structure. To achieve persistent silencing effects, it may be necessary to introduce a plasmid construct that can continually express the GSA in the host cells. Our study also demonstrated the synergistic effect of multiple ASOs in silencing the *PDS* gene, indicating that multiple GSAs may be required to silence certain genes. A single construct expressing multiple GSAs has been reported to silence multiple genes in thale cress (*Arabidopsis thaliana* [98]). A similar strategy may be applied in future studies with *P. australis* by introducing a construct expressing multiple GSAs to target a single or multiple essential gene(s) simultaneously to facilitate the higher efficacy of gene silencing and enhanced phenotypic alteration in this invasive species of concern. To address such issues as the presence of multiple *PDS* paralogs and splice variants in *P. australis*, we may have to design more broad-spectrum GSAs that target all PDS paralogs and variants to silence these genes and induce phenotypic changes.

GSA-mediated gene silencing provides an innovative and promising opportunity for developing species-specific, ecofriendly, and sprayable bioherbicides to control invasive plant species and weeds [99]. However, technological development still faces multiple challenges, including the lack of well-annotated genomes for the species of concern; the identification and characterization of target gene candidates; the design and screening of highly potent GSAs that can induce unrepairable damage if overwhelming their target genes; the efficient delivery, internalization, stability, and systematic movement of GSAs via a viral vector or non-vial nanocarrier; and the scalable production of GSAs and nanocarriers. Despite these roadblocks, science is advancing rapidly, and individual challenges are being resolved. For instance, the reference genomes and other genomic resources of 27 weed species have been made publicly available for researchers by the International Weed Genomics Consortium (IWGC) [100]. Besides CPP, other nanocarriers, such as LDH, SWCNT, MWCNT, and CND, have been applied successfully to plants for GSA delivery [42,43,44,45,46]. More recently, Cai et al. [101] used mesoporous silica nanoparticles for foliar spraying siRNAs targeting *PDS* and magnesium chelatase genes and observed both gene silencing and the photobleaching phenotype. Pal et al. [102] resolved the stability issue for dsRNA by encapsulating it in a cationic poly-aspartic acid-derived polymer (CPP6) and delivered a PDS-targeting dsRNA-CPP6 complex to rice through root uptake, which effectively silenced the *PDS* gene and induced a dwarf and albino phenotype. With the recent commercial sale of Calantha^TM^ (Greenlight Biosciences Inc., Medford, MA, USA), an RNAi-based dsRNA spray pesticide killing Colorado potato beetles [103], the gene-silencing-based mode of action could be similarly harnessed to develop next-generation herbicides that would become an indispensable and safe tool for weed management and invasive plant control.

## 5. Conclusions

Here, we report for the first time the use of CPPs as the vehicle to deliver several types of nucleic acids or constructs that suppress the expression of a putative endogenous *PDS* gene in the invasive subspecies *P. australis* ssp. *australis* via folia application. These nucleic acid-based GSAs, including dsRNA, amiRNA, and ASO, or GSA-expressing constructs induced post-transcriptional gene silencing in vivo. We also presented microscopic evidence supporting the internalization of fluorescence-labeled CPP–GSA complexes. Although there was a lack of physiological trait alteration (i.e., bleaching phenotype in treated plants), our study demonstrated the CPP-mediated delivery of GSAs and the knockdown of their targeted endogenous gene in a non-model, invasive monocotyledonous plant species. Our work lays a foundation for further research toward developing convenient, cost-effective, field-deployable, and ecofriendly gene-silencing technologies for invasive weed management.

## Figures and Tables

**Figure 1 plants-14-00458-f001:**
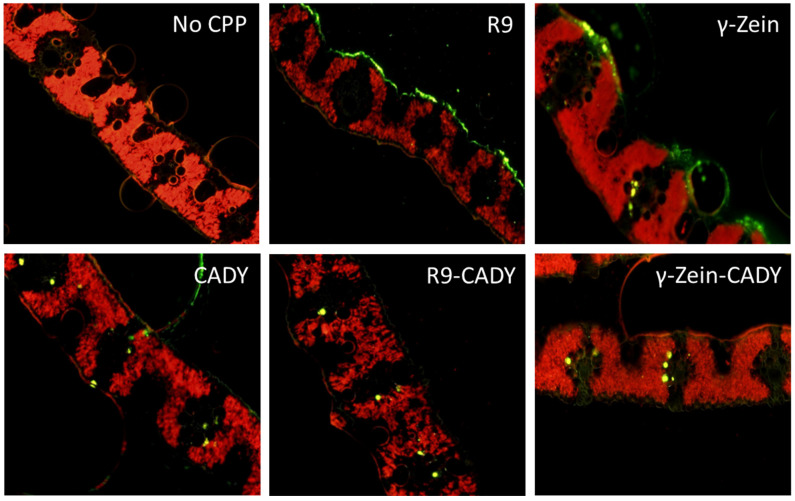
Ex vivo penetration capability of five cell-penetrating peptides (CPPs) into freshly cut *P. australis* leaf. The CPPs are R9, γ-zein, CADY, R9-CADY, and γ-zein-CADY, all labeled with FITC. Shown are fluorescent field images of leaf tissue cross sections at 10× magnifying power using a Leica DM IL LED inverted fluorescence microscope. The cross sections were made one hour after the foliar application of individual CPPs at a rate of 5 nmol of CPP dissolved in 50 µL of infiltration medium per leaf to the adaxial (upper) surface of leaves. Yellow fluorescence indicates FITC-labeled CPP signals, and red fluorescence indicates chlorophyll.

**Figure 2 plants-14-00458-f002:**
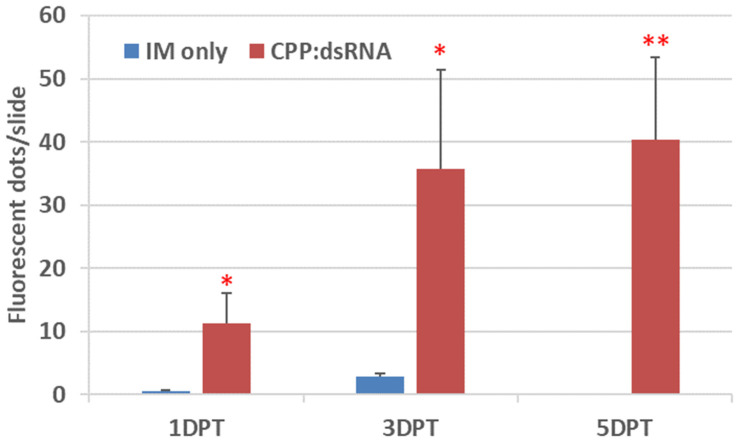
Quantitative measurement of internalized FAM-dsRNA*_PDS_* as a payload of γ-zein-CADY. The FAM-labeled dsRNA*_PDS_* complex with non-labeled γ-zein-CADY (charge ratio = 1:1) was applied in vivo to *P. australis* leaf surfaces at a rate of 1 nmol of dsRNA/leaf. The CPP:dsRNA-treated and the infiltration medium (IM)-only control leaves were sampled at 1, 3, and 5 days post treatment (DPT). Green fluorescent (FAM) dots were counted for an intact cross section on each slide, three slides per leaf, and three leaves per treatment. Shown are the mean (column) + standard error (bar) of fluorescent dots/slide (n = 9). Two-tailed *t*-tests were conducted to infer statistical significance between the control and the treated leaves, with “*” denoting 0.01 < *p* < 0.05 and “**” denoting 0.001 < *p* < 0.01.

**Figure 3 plants-14-00458-f003:**
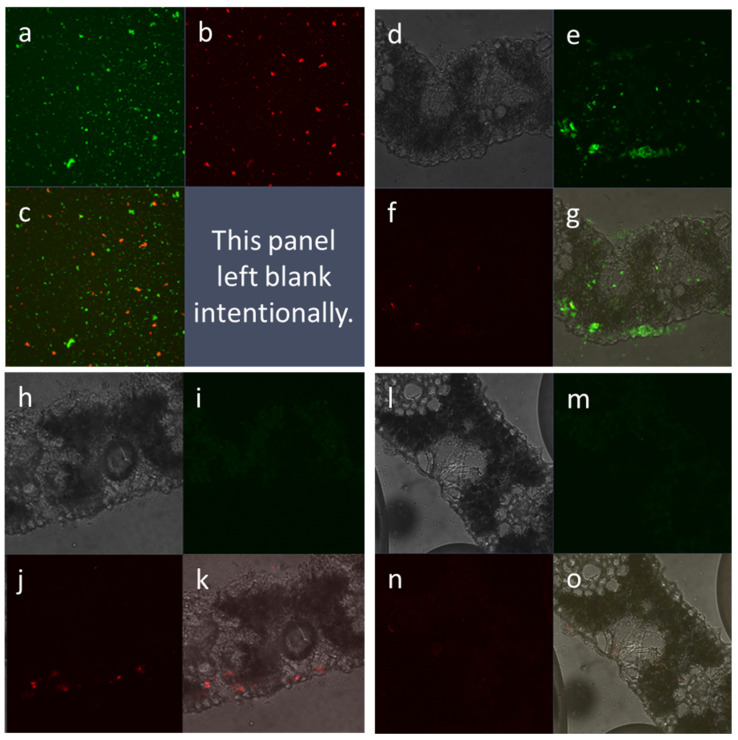
Cellular internalization of Atto550-labeled amiDNA*_PDS-1_* as a cargo of FITC-labeled (**d**–**g**) or non-labeled (**h**–**k**) γ-zein-CADY. The CPP and GSA were mixed at a charge ratio of 1:50 except for (**h**–**k**) at 1:6.25 (GSA:CPP) (GSA:CPP). The GSA:CPP complex was applied in vivo to *P. australis* leaf surfaces 4 days before sampling for cross-section slide preparation. The foliar application solution of Atto550-amiDNA*_PDS-1_*: FITC-γ-zein-CADY served as the positive control (**a**–**c**). The Atto550-labeled amiDNA*_PDS-1_*-only application (**l**–**o**) served as the negative control. Images of the blank control (infiltration medium only without GSA and CPP) are not shown. The cross-section images were taken at 63× magnification using a Zeiss LSM 510 META confocal microscope. (**a**,**e**,**i**,**m**) green (FITC) field; (**b**,**f**,**j**,**n**) red (Atto550) field; (**c**) merged green and red fields; (**d**,**h**,**l**) bright field; (**g**,**k**,**o**) merged bright, green, and red fields.

**Figure 4 plants-14-00458-f004:**
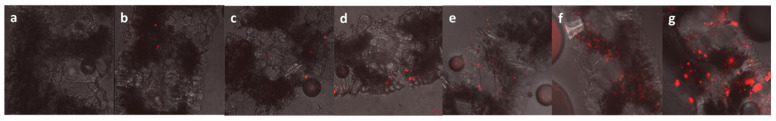
Confocal microscopic pictures of cross-section slides prepared from *P. australis* leaves treated for 4 days with an Atto550-labeled-amiDNA*_PDS-1_*:γ-zein-CADY complex at different CPP:GSA charge ratios. (**a**) Control (CPP only); (**b**) 25:1; (**c**) 12.5:1; (**d**) 6.25:1; (**e**) 4:1; (**f**) 2:1; (**g**) 1:1. Control samples were used to filter out background autofluorescence, and the same settings were used across all samples (e.g., magnification, laser intensity, gain of ~500). The red dots represent the fluorescence-labeled CPP-GSA complex.

**Figure 5 plants-14-00458-f005:**
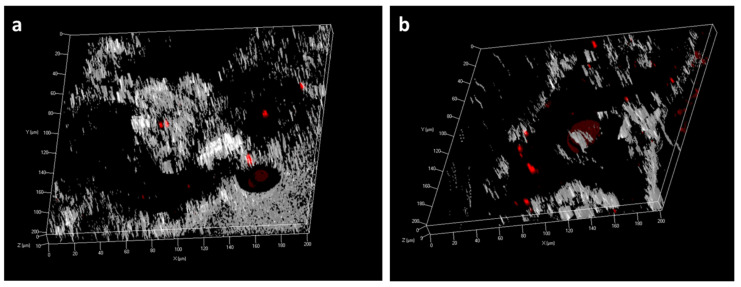
The z-stack confocal images of the Atto550-amiDNA*_PDS-1_*:γ-zein-CADY complex-treated *P. australis* leaves sampled at 4 days post treatment. The two different Atto550-amiDNA*_PDS-1_*:γ-zein-CADY charge ratios are (**a**) 1:12.5 and (**b**) 1:6.25. Z-stack-images were collected at 1–2 µm intervals, and a 3D rendering was generated for a greater depth of field. The red dots represent the fluorescence-labeled CPP-GSA complex.

**Figure 6 plants-14-00458-f006:**
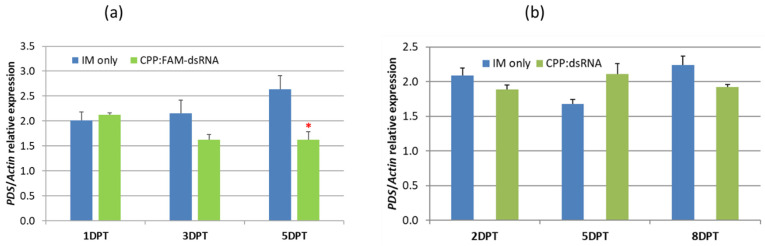
In vivo effects of γ-zein-CADY:dsRNA*_PDS_* on the expression of the putative *PDS* gene in treated *P. australis* leaves. The CPP to GSA charge ratio was 1 (γ-zein-CADY):1 (1 nmol dsRNA*_PDS_*), and a foliar solution of 50 µL (containing CPP:GSA complex or IM only) was applied to each leaf. The infiltration medium (IM)-only solution served as the control. Two experiments were conducted with dsRNA*_PDS_* as the GSA. In the first experiment (**a**), dsRNA*_PDS_* was labeled with FAM (also refer to Figure 2 and Methods), and treatment lasted for up to 5 days. In the second experiment (**b**), dsRNA*_PDS_* was non-labeled, and treatment lasted for up to 8 days. Mean = column; error bar = standard deviation; “*” denotes statistical significance at 0.01 < *p* < 0.05 (2-tailed *t*-test, n = 3). *Actin* was selected as the reference gene for the relative quantification of *PDS* gene expression.

**Figure 7 plants-14-00458-f007:**
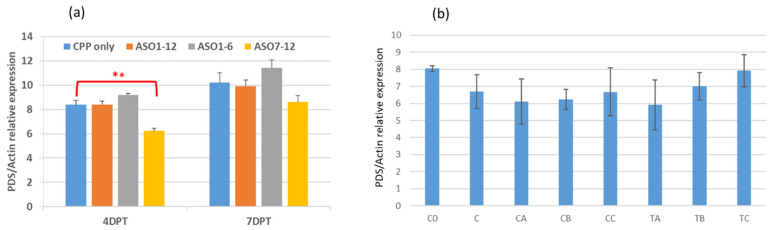
In vivo effects of ASO:γ-zein-CADY complex on the expression of putative *PDS* relative to *actin* in *P. australis* leaves treated for 4 or 7 days. DPT = days post treatment. Charge ratio = 1:6.25 (ASO:CPP). The ASO:CPP application rate was 1 nmol ASO pool and 5.4 nmol of CPP per leaf. Each pool of ASOs was made up of equal amounts of individual components. Shown are the mean (column) and standard error (bar), with n = 6. “**” denotes statistical significance at *p* < 0.01 (ANOVA with Tukey post hoc test, n = 6). In (**a**) (**left**), CPP only = γ-zein-CADY (CPP) + infiltration medium (IM); ASO1–12 = pool A (ASO-1 to ASO-12) + CPP + IM; ASO1–6 = pool B (ASO-1 to ASO-6) + CPP + IM; ASO7–12 = pool C (ASO-7 to ASO-12) + CPP + IM. In (**b**) (**right**), C0 = blank control at day 0; C = blank control at day 4; CA = (ASO7–12 + IM) control; CB = IM control; CC = (CPP + IM) control; TA = (ASO-7 to ASO-12) + CPP + IM; TB = (ASO-7 to ASO-9) + CPP + IM; TC = (ASO-10 to ASO-12) + CPP + IM; except for C0, all treatments were sampled at 4 days post treatment (DPT).

**Figure 8 plants-14-00458-f008:**
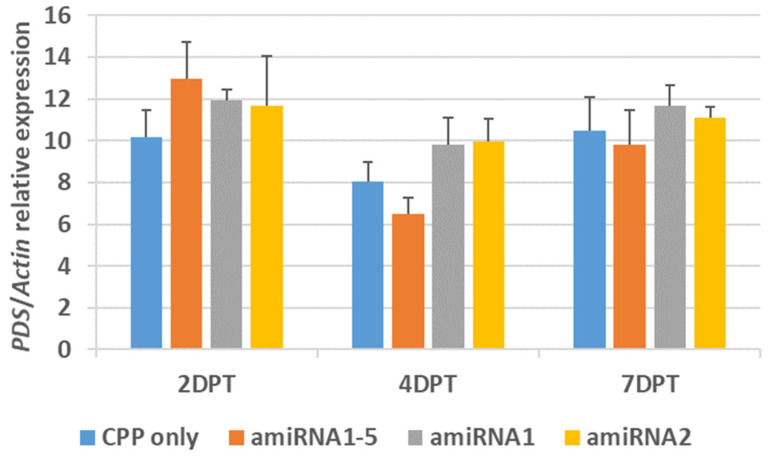
In vivo effects of γ-zein-CADY: amiRNA*_PDS_* complexes on the expression of putative *PDS* relative to *actin* in treated *P. australis* leaves for up to 7 days post treatment (DPT). The CPP to GSA charge ratio was 12.5 (γ-zein-CADY):1 (1 nmol amiRNA*_PDS_*), and a foliar solution of 50 µL (containing CPP:amiRNA complex or CPP only and infiltration medium (IM)) was applied to each leaf. The CPP-only (+IM) solution served as the control. The three CPP:amiRNA treatments include amiRNA1–5 (i.e., equal amount for each individual amiRNA*_PDS_*), amiRNA1 (amiRNA*_PDS-1_* only), and amiRNA2 (amiRNA*_PDS-2_* only) (refer to Table 3 for amiRNA*_PDS_* sequences). No statistically significant difference was observable between the CPP-only control and each amiRNA treatment (2-tailed *t*-test, n = 3, *p* >0.05). *Actin* was selected as the reference gene for the relative quantification of *PDS* gene expression.

**Figure 9 plants-14-00458-f009:**
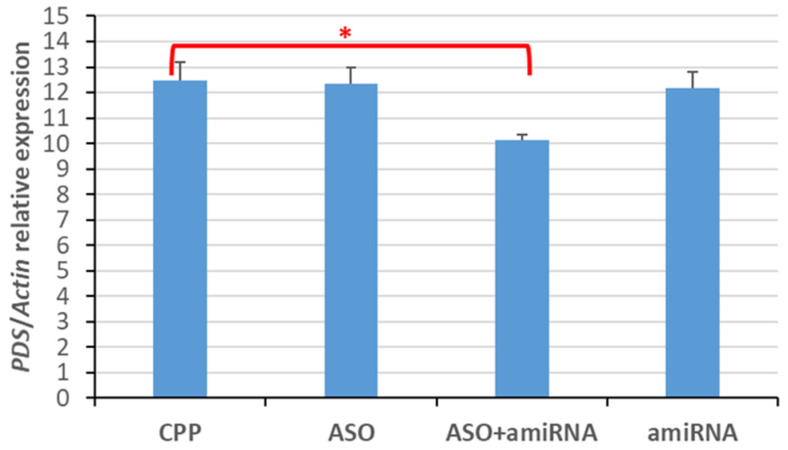
In vivo effects of pooled amiRNAs and/or pooled ASOs on the expression of the putative *PDS* gene in *P. australis* leaves treated daily for 4 days. The GSAs (amiRNAs/ASOs) were delivered by γ-zein-CADY (CPP). Charge ratio = 1:2 (GSA:CPP). The GSA:CPP application rate was 1 nmol of GSA and 5.4 nmol of CPP per leaf. Shown are the mean (column) + standard error (bar), with n = 6. “*” denotes statistical significance at 0.01 < *p* < 0.05 (2-tailed *t*-test, n = 6). Four treatments: CPP = γ-zein-CADY (CPP) + infiltration medium (IM); ASO = pool C of ASO-7 to ASO-12 + CPP + IM; ASO + amiRNA = pool C of ASO-7 to ASO-12 or a pool of amiRNA*_PDS-1_* to amiRNA*_PDS-5_* + CPP + IM; amiRNA = pool of amiRNA*_PDS-1_* to amiRNA*_PDS-5_* + CPP + IM. Each pool was made up of equal amounts of individual components. The ASO + amiRNA treatment was performed by alternating daily applications of ASO:CPP and amiRNA:CPP (i.e., ASOs, amiRNAs, ASOs, and amiRNAs on day 1, 2, 3, and 4, separately).

**Table 1 plants-14-00458-t001:** Documented cell-penetrating peptides (CPPs) used for gene delivery into plant cells.

CPP	Amino Acid Sequence	Origin	Classification	Charge	Plant Cell	Reference
γ-Zein	VRLPPPVRLPPPVRLPPP	γ-zein storage protein from maize	Proline-rich	(+3)	Protoplast	[49,70]
Bac7	RRIRPRPPRLPRPRPRPLPFPRPG	natural protegrin bactenecin family	Proline-rich	(+9)	Unknown	[71]
Bp100	KKLFKKILKYL	natural protegrin from *Hyalophora cecropia*	Cationic	(+5)	Living tobacco cells	[66]
SV40 NLS	PKKKRKV	simian virus 40	Cationic	(+5)	Plant nuclei	[72]
Penetratin	RQIKIWFQNRRMKWKK	natural protegrin *Drosophila*-antennapedia	Cationic	(+7)	Protoplast	[64]
Tachyplesin	KWCFRVCYRGICYRRCRGK	marine antimicrobial peptides	Cationic	(+7)	Root tip, coleoptile & hypocotyl	[67]
Tat	YGRKKRRQRRR	HIV-1 Tat protein	Cationic	(+8)	Root tip, epidermal & mesophyll protoplast	[58,59,62]
Polyarginine (R9)	RRRRRRRRR	synthetic	Cationic	(+9)	Root tip & epidermal	[58,59,60,61]
Polyarginine (R12)	RRRRRRRRRRRR	synthetic	Cationic	(+12)	Suspension cells	[65]
(KH)_9_-Bp100	KHKHKHKHKHKHKHKHKHKKLFKKILKYL	fusion of synthetic (KH)_9_ and natural Bp100	Cationic	(+23)	Intact leaf cells of *Arabidopsis thaliana*	[56]
R9-Bp100, (KH)_9_-Bp100, R9-Tat_2_	Refer to above	refer to above for components	Cationic	varied	Intact leaf cells	[55]
Bp100, K_8_, (KH)_9_, (Bp100)_2_K_8_, Bp100(KH)_9_	Refer to above for Bp100 and (KH)_9_ except K_8_ = KKKKKKKK	refer to above for components except K_8_ being synthetic	Cationic	varied	Intact leaves	[68]
Transportan	GWTLNSAGYLLGKINLKALAALAKKIL	neuropeptide galanin and mastoparan chimeria	Bipartite/chimeric	(+4)	Protoplast	[63,64,73]
pVEC	LLIILRRRIRKQAHAHSK	murine vascular endothelial cadherin	Bipartite/chimeric	(+8)	Protoplast	[63,64]
CADY	GLWRALWRLLRSLWRLLWRA	synthetic nucleic acid binding peptide	Amphipathic	(+5)	Unknown	[74]
SynB1	RGGRLSYSRRRFSTSTGR	porcine protegrin-1	Amphipathic	(+6)	Unknown	[75]

**Table 2 plants-14-00458-t002:** Amino acid sequence and length of five new fusion cell-penetrating peptides (CPPs) designed and used in the present study, following the scheme “CPP1–spacer–(CPP2–spacer)–EED”, with EED being an endosomal escape domain. Some amino acids in the fusion CPPs are color coded to highlight the spacer (in green font and underlined) and the EED (in red font).

CPP Name	Peptide Sequence (N-Terminus → C-Terminus)	Length (aa)
γ-Zein	VRLPPPVRLPPPVRLPPPGGFWFG	24
R9	RRRRRRRRRGGFWFG	15
CADY	GLWRALWRLLRSLWRLLWKGGFWFG	25
γ-Zein-CADY	VRLPPPVRLPPPVRLPPPGGLWRALWRLLRSLWRLLWKGGFWFG	44
R9-CADY	RRRRRRRRRGGLWRALWRLLRSLWRLLWKGGFWFG	35

**Table 3 plants-14-00458-t003:** Two classes of gene-silencing agents (GSAs) designed to target a putative *PaPDS* (*P. australis* phytoene desaturase) gene.

**GSA Class**	**GSA Name**	**Nucleic Acid Sequence (5′** **→ 3′)**
ASO	ASO-1	T*T*C*AGTTTCACTTCGTCC*A*A*C
ASO	ASO-2	T*G*A*TTTCAGTTTCACTTC*G*T*C
ASO	ASO-3	T*A*G*CTCTTCCATAGTTGC*A*T*C
ASO	ASO-4	T*G*C*TAGCTCTTCCATAGT*T*G*C
ASO	ASO-5	A*C*T*TTGATCGGCAGCAAT*T*T*C
ASO	ASO-6	A*A*C*AGATCTCGGTGTCTT*C*A*C
ASO	ASO-7	T*T*G*TAAACAGATCTCGGT*G*T*C
ASO	ASO-8	A*T*T*TCTGCTTCGTGTAAT*C*G*C
ASO	ASO-9	T*T*G*TAAAGGTCCTCCTCC*T*A*C
ASO	ASO-10	A*A*C*TCAGTCACAATTCAA*C*T*C
ASO	ASO-11	A*T*A*TCAACTCAGTCACAA*T*T*C
ASO	ASO-12	A*A*T*ATCTGCGATCTCTTT*A*T*C
amiRNA	amiRNA*_PDS-_*_1_	UAUUCUUUUAAUGAGAGGCGG
amiRNA	amiRNA*_PDS-_*_2_	UUGGUAAAUAUACUAUGCCCC
amiRNA	amiRNA*_PDS-_*_3_	UAUAUUAUCCUAAGACGUCGG
amiRNA	amiRNA*_PDS-_*_4_	UUGGUAAAUAUACUAUGGCCA
amiRNA	amiRNA*_PDS-_*_5_	UUUUAAUGAGAGACGGACCAG
Atto550-amiDNA	amiDNA*_PDS-_*_1_	TATTCTTTTAATGAGAGGCGG

Note: “*” denotes phosphorothioate internucleotide linkage.

**Table 4 plants-14-00458-t004:** qPCR primer sequences.

Primer	Sequence (5′-3′)	Note
Actin183 F	GGATGATATGGAGAAGATCT	Reference gene (*Actin*)
Actin307 R	GGGTCATCTTCTCACTATTA	
PDS95F	GGAGTTCACTTTTGAGTG	Target gene (*PDS*)
PDS251R	GCTAGCTCTTCCATAGTT	

## Data Availability

Data are contained within the article and Appendix A. The raw qPCR data are available online at https://doi.org/10.5066/P13ETPDI.

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
