# Peer review of "Cell-Penetrating Peptide-Mediated Delivery of Gene-Silencing Nucleic Acids to the Invasive Common Reed *Phragmites australis* via Foliar Application"

_plants, 2025, doi:10.3390/plants14030458_

Round 1

Reviewer 1 Report

Comments and Suggestions for Authors

This study demonstrates the use of cell-penetrating peptides (CPPs), specifically γ-zein-CADY, to deliver gene silencing agents into monocot Phragmites leaf tissue for targeted PDS gene knockdown.

1. The study addresses the delivery challenge of GSA, focusing primarily on the issue of sustained release. Has any experiment been conducted to assess how long CPP can sustain GSA in both in-situ and ex-situ conditions?

2. Have you verified whether these CPPs are releasing effectively, or if they might inhibit RNAi activity on the conjugate?

3. Ideally, dsRNA should produce a stronger silencing effect than a group of amiRNAs or ASOs, as dsRNA contains more siRNA target sites.

4. In the CPP-dsRNA conjugation, some unbound dsRNA may remain. Have you removed the unbound dsRNA before proceeding with the experiments?

Results 3.1:

5. The use of FITC Labelling provides clear visual evidence, further quantitative measurement of the fluorescence will give more clarity, if you have done it can be added to the supplementary file

6. Whether the freshly cut leaves in the ex-vivo set up could impact CPP uptake differently compared to intact in-vivo leaves

7. Two different unit were used for the concentration of CPP which is little confusing, same unit should be used in legend of Fig 1 and result section 3.1

Results 3.2:

8. Dual Fluorescence labelling (FITC for CPP and AJO 550 for amiDNA) is an effective method to visualize confirm complex formation

9. Outline the basis for the original 1:1 and 50:1 ratio for dsRNA & amiDNA

Results 3.3:

10. Certain phrases, such as “sitting on top or below the leaf surface” and “a large amount of the fluorescent signal,” are somewhat informal. Revising these phrases with precise language, such as “localized on the surface” or “a significant amount of the fluorescence signal,” would improve the text’s professionalism and clarity.

11. The text notes that the amiDNAPDS-1 signal was most abundant at the 25:1 ratio, with much of the fluorescent signal outside the leaf tissue. While it suggests a “saturating concentration,” it does not clarify if this observation means that beyond a certain concentration, further increases in CPP amount result in inefficiency or excess deposition outside the tissue.

12. Explain why amiRNAPDS-1 and amiRNAPDS-2 were chosen for single treatments over the other amiRNAs. Was this selection based on prior data or sequence specificity?

13. Certain terms, like "significant reduction in relative PDS gene expression," could be clarified to read as "statistically significant reduction in PDS gene expression" for precision.

Results 3.4:

14. In both the experiment FAM labelling dsRNA used? Does FAM labelling consistently protect dsRNA from nuclease degradation or this was unexpected?

15. Phrase such as “significant reduction in transcription gene expression” could be more precise. Fold change can be in-cooperated

16. Rephrasing "alternation between CPP + ASO Pool C and CPP + amiRNA pool" as "alternating applications of CPP + ASO Pool C and CPP + amiRNA pool treatments" could improve clarity.

17. The process of creating three ASO pools have any rational behind it?

18. Pool A includes ASO 1 through ASO 6 while pool C contains ASO7 through ASO 12 restate the details, currently this information is split between the sentences which make it confusing

19. Why amiRNA PDS-1 and amiRNA PDS-2 were chosen for single treatment clarify

20. Some phrase can be reconstructed, for example instead of significant reduction in target gene expression use significant reduction in PDS gene, similarly unstable complexation could be rephrased as inefficient or unstable binding between CPP and GSA for clarity.

Author Response

Comment 1. The study addresses the delivery challenge of GSA, focusing primarily on the issue of sustained release. Has any experiment been conducted to assess how long CPP can sustain GSA in both in-situ and ex-situ conditions?

Response 1: This is a very good question. No experiment was conducted to address this question in the present study because we had not identified a GSA that not only knocks down the PDS gene but also induces visible phenotypic changes. We are also unaware of any other published work that studied this question. For future studies, we plan to further investigate over a time course the stability and sustainability of CPPs (or any other nanocarriers) and the release/mobility of a GSA from its conjugation with a CPP (or other nanocarriers) once the GSA-CPP complex has reached its destination. We’d also like to bring to the reviewer’s attention that we added an endosomal escape domain (EED) to the in-house designed fusion and non-fusion CPPs. It is widely accepted that as intracellular sorting organelles, endosomes in plant cells regulate trafficking of proteins and lipids among other subcellular compartments of the secretory and endocytic pathway, specifically the plasma membrane Golgi, trans-Golgi network (TGN), and vacuoles or lysosomes (see https://www.nature.com/scitable/topicpage/endosomes-in-plants-14404958/). The incorporation of an EED allows an internalized CPP through endocytosis to escape from being trapped in endosomes and transported into or sequestered within subcellular compartments. Instead, the CPP containing an EED (along with a GSA cargo) could be released into the cytoplasm (https://www.nature.com/articles/srep32301) so that the GSA could hybridize to and suppress the transcripts expressed by its target gene. Therefore, we believe that the addition of EED to CPP designs can improve the sustainability of CPPs and the delivery efficiency of the GSA payload.

Comment 2. Have you verified whether these CPPs are releasing effectively, or if they might inhibit RNAi activity on the conjugate?

Response 2: This is one of the topics for our future research as mentioned above in Response 1. It is worth a separate full-scale effort to verify the releasing of GSA from CPP upon the arrival of the CPP-GSA complex at the target subcellular compartments. Meanwhile, it has been well documented that many CPPs (e.g., R9, γ-zein and CADY) are capable of penetrating cell membranes and delivering the GSA cargos inside plant cells. We are unaware of any studies reporting inhibitory effects of CPPs on GSA activity (RNAi) in plants. We did find a study where the gene-silencing efficiency of covalently conjugated PF28-TG1 was significantly lower than that of the non-covalently complexed PF14:TG1, as PF28 (a CPP) probably shielded the interaction between the payload peptide (TG1) and its target U87 (human glioblastoma cell) MG receptor or hindered siRNA release into the cytosol (see https://doi.org/10.1016/j.peptides.2018.04.015). That study suggests that non-covalent binding between a CPP and a GSA may be a more efficient strategy for GSA delivery. In addition, we also noticed a recent study where the authors evaluated the impact of payload conjugation on the cell-penetrating efficiency of the endosomal escape peptide dfTAT (see https://pubs.acs.org/doi/10.1021/acs.bioconjchem.3c00369). The study demonstrated that payload size and property (e.g., GSA cargo type and other physiochemical features) affected GSA delivery efficiency. That was why we compared different CPPs and GSA types and investigated the GSA:CPP ratio to improve delivery efficiency. In future studies, we will optimize the RNAi activity of GSA by screening various delivery vehicles (CPPs, nanocarriers, etc.) once we have identified an effective GSA.

Comment 3. Ideally, dsRNA should produce a stronger silencing effect than a group of amiRNAs or ASOs, as dsRNA contains more siRNA target sites.

Response 3: We agree with the reviewer on this point. However, given that only an incomplete PDS gene sequence (i.e., a 788-bp transcript) was available when the present study was conducted, we were only able to design one dsRNA and tested it for gene silencing effects. The work reported in this manuscript represents our early foundational efforts in developing RNAi-based gene silencing technologies for invasive plant species control. Given so many constrains we faced with a poorly genetically characterized non-model organism, progress would be made gradually as we learn more about the basic genetic and genomic characterizations and functional genomics for the plant species of interest, i.e., the invasive common reed subspecies Phragmites australis subsp. australis. For instance, there are now 6 genome assemblies deposited in the NCBI databases (see https://www.ncbi.nlm.nih.gov/datasets/genome/?taxon=29695). We have identified two PDS paralogs and designed and screened multiple dsRNAs to identify the most effective one. Further work is undergoing and new results will be prepared and submitted for publication soon.

Comment 4. In the CPP-dsRNA conjugation, some unbound dsRNA may remain. Have you removed the unbound dsRNA before proceeding with the experiments?

Response 4: No, we did not remove the unbound dsRNA before applying the γ-zein-CADY:dsRNA conjugates to the surface of Phragmites leaves because we thought that the removal would not affect our experimental results.

Comment 5. The use of FITC Labelling provides clear visual evidence, further quantitative measurement of the fluorescence will give more clarity, if you have done it can be added to the supplementary file.

Response 5: We used FITC-labelled CPPs and FAM- or Atto550-labelled GSA to investigate CPP-mediated GSA delivery in both qualitative and semi-quantitative fashions. The key findings are presented in Figures 1-5. Figure 1 shows that FITC-labelled CPPs can enter leaf tissues within as short as 1 hour. Figures 2 & 3 demonstrate that it might take 3-5 days for γ-zein-CADY to deliver all its payload to intracellular compartments at various CPP:GSA charge ratios (ranging from 50:1 to 1:1). Figure 4 shows semi-quantitatively that the amount of internalized amiRNA increased as the CPP:GSA ratio decreased from 50:1 to 1:1, and by adjusting the charge ratio between 12.5:1 and 2:1 one should be able to deliver the desired amount of GSA in a semiquantitative fashion. Figure 5 provides further evidence for the 3-D localization of the delivered and internalized GSA. We believe that these pieces of graphical and numerical information provide evidence that suffices the need for demonstrating effective delivery of GSAs via our redesigned fusion or non-fusion CPPs.

Comment 6. Whether the freshly cut leaves in the ex-vivo set up could impact CPP uptake differently compared to intact in-vivo leaves

Response 6: We did not pursue this line of research as we used the 1-hour, ex-vivo treatment to rapidly screen the uptake efficiency of 5 in-house designed CPPs. We assumed that if a CPP could be taken up ex vivo, it should behave similarly in vivo.

Comment 7. Two different unit were used for the concentration of CPP which is little confusing, same unit should be used in legend of Fig 1 and result section 3.1.

Response 7: We very much thank the reviewer for catching this error. The unit used for expressing the amount of applied CPP should have been nmol/leaf, not µg/leaf. We have corrected the error in line 313 on page 17 of the revised manuscript.

Comment 8. Dual Fluorescence labelling (FITC for CPP and AJO 550 for amiDNA) is an effective method to visualize confirm complex formation.

Response 8: We totally agree with the reviewer on this. Our results also support this viewpoint.

Comment 9. Outline the basis for the original 1:1 and 50:1 ratio for dsRNA & amiDNA

Response 9: We accepted this suggestion and added a few sentences to explain why we chose the CPP:GSA charge ratio ranging between 1:1 and 50:1. See lines 335-339 on page 18.

Comment 10. Certain phrases, such as “sitting on top or below the leaf surface” and “a large amount of the fluorescent signal,” are somewhat informal. Revising these phrases with precise language, such as “localized on the surface” or “a significant amount of the fluorescence signal,” would improve the text’s professionalism and clarity.

Response 10: We accepted the suggested changes and revised the phrases accordingly. See lines 400-401 on page 22.

Comment 11. The text notes that the amiDNAPDS-1 signal was most abundant at the 25:1 ratio, with much of the fluorescent signal outside the leaf tissue. While it suggests a “saturating concentration,” it does not clarify if this observation means that beyond a certain concentration, further increases in CPP amount result in inefficiency or excess deposition outside the tissue.

Response 11: First, we’d like to apologize for mistakenly reversing the charge ratios between CPP and GSA in Figure 4 (i.e., the CPP:GSA charge ratios should have been 25:1 to 1:1, instead of 1:1 to 1:25), leading to wrong interpretation of the images in panels 4b to 4g. Second, the CPP amount used was set at a fixed level (ca. 5 nmol/leaf) based on our preliminary experiments and the ratio between CPP and GSA was adjusted by mixing the fixed amount of CPP with varying amounts of GSA. Therefore, this experiment was intended to determine the “optimal” or appropriate GSA loading capacity of CPP. Results suggest that 2:1 (CPP:GSA) appeared to be the maximum loading capacity of γ-zein-CADY with amiDNAPDS-1. We revised the entire paragraph to correct the CPP:GSA charge ratios and interpretation of Figure 4 (including the figure legend). See lines 392-408 on page 22 and lines 413-414 and 420 on page 23.

Comment 12. Explain why amiRNAPDS-1 and amiRNAPDS-2 were chosen for single treatments over the other amiRNAs. Was this selection based on prior data or sequence specificity?

Response 12: We selected these two amiRNAs because they ranked the highest among the top five amiRNAs selected for testing. We added a sentence to explain the choice (see lines 477-478 on page 26).

Comment 13. Certain terms, like "significant reduction in relative PDS gene expression," could be clarified to read as "statistically significant reduction in PDS gene expression" for precision.

Response 13: We accepted this suggestion and revised accordingly at three instances (page 23 line 425, page 25 line 454, and page 26 line 473).

Comment 14. In both the experiments (where) FAM labelling dsRNA used, does FAM labelling consistently protect dsRNA from nuclease degradation or this was unexpected?

Response 14: We presented only one experiment that involved FAM-labelled dsRNA. In this experiment, we determined both CPP-mediated dsRNA uptake (Figure 2) and dsRNA effects on target PDS gene expression (Figure 6a). In the second dsRNA experiment, we used non-labelled dsRNA complexed with γ-zein-CADY (Figure 6b). We attributed the difference between the two dsRNA experiments to FAM labelling that protected dsRNA from nuclease degradation, which was unexpected. Stability of dsRNA (i.e., resistance against intracellular nuclease degradation) is obviously a key issue and big challenge we need to resolve down the road.

Comment 15. Phrase such as “significant reduction in transcription gene expression” could be more precise. Fold change can be incorporated.

Response 15: We modified the phrase as suggested and added “38%” to specify the degree of expression suppression. The revised sentence (page 23 lines 425-426) now reads “a statistically significant 38% reduction in transcriptional gene expression (p < 0.05) was observed…”.

Comment 16. Rephrasing "alternation between CPP + ASO Pool C and CPP + amiRNA pool" as "alternating applications of CPP + ASO Pool C and CPP + amiRNA pool treatments" could improve clarity.

Response 16: We slightly changed the phrase into “alternation of CPP + ASO Pool C and CPP + amiRNA pool”. See page 28 line 505. We consulted Oxford Languages, Google’s English dictionary, for the definition of “alternation”, which is “the repeated occurrence of two things in turn”. If looking at the context “daily treatment for 4 consecutive days with CPP (γ-zein-CADY) alone, CPP + ASO Pool C (ASO-7 to ASO-12), CPP + amiRNA pool (amiRNAPDS-1 to amiRNAPDS-5), or alternation of CPP + ASO Pool C and CPP + amiRNA pool” and referring to Figure 9 (and figure legend), there should be no confusion for the four treatments in our experimental design. And the fourth treatment was ASOs, amiRNAs, ASOs, and amiRNAs on day 1, 2, 3, and 4, separately.

Comment 17. The process of creating three ASO pools have any rational behind it?

Response 17: We created Pools B and C in considerations of reducing the dilution factor (1/12 in Pool A vs 1/6 in Pools B and C) for each constituent ASO and narrowing down the effective ASO candidates (from 12 to 6). We added this rationale in the revised manuscript. See page 24 lines 445-447.

Comment 18. Pool A includes ASO 1 through ASO 6 while pool C contains ASO7 through ASO 12 restate the details, currently this information is split between the sentences which make it confusing.

Response 18: There are initially three ASO pools. Pool A is made up of all 12 ASOs, i.e., ASO-1 to ASO-12. Pools B and C each contains 6 ASOs, i.e., ASO-1 to ASO-6 in Pool B and ASO-7 to ASO-12 in Pool C. This information is presented in both the text (page 24 lines 444-445) and the legends of Figures 7a and 9. We further split Pool C into two sub-treatment groups: TB (ASO-7 to ASO-9) and TC (ASO-10 to ASO-12), with TA being the same as Pool C (see page 25 lines 449-451 and Figure 7b legend).

Comment 19. Clarify why amiRNAPDS-1 and amiRNAPDS-2 were chosen for single treatment.

Response 19: This is identical to Comment 12. Please see Response 12 above.

Comment 20. Some phrase can be reconstructed, for example instead of significant reduction in target gene expression use significant reduction in PDS gene, similarly unstable complexation could be rephrased as inefficient or unstable binding between CPP and GSA for clarity.

Response 20: We appreciate the suggestions. As recommended, we changed “target gene expression” into “target PDS gene expression” at three instances (page 16 line 298, page 25 line 454, and page 31 line 597), and “unstable complexation” into “inefficient or unstable binding” in line 456 on page 25.

Reviewer 2 Report

Comments and Suggestions for Authors

The study by Ji et al. represents the first application of cell-penetrating peptides (CPPs) to deliver gene-silencing agents (GSAs) in a monocot plant for controlling the invasive species Phragmites australis. The authors combined CPPs with the EED (Endosomal Escape Domain) sequence (GFWFG) to enhance intracellular delivery, preventing endosomal trapping and degradation by lysosomal enzymes. This study generated extensive data, presented the results clearly, and provided acceptable figures and tables. However, this reviewer has the following suggestions and concerns:

  1. Control in Figure 2:
    The lack of an appropriate control in Figure 2 is a concern. A negative control, such as FAM-labeled dsRNA without CPP, should be included when counting the number of fluorescent dots. This would confirm whether surface proteins have been effectively removed. Additionally, the authors used a 10% phosphate buffer to wash the leaves, which removes only loosely bound surface proteins. For complete removal of surface-bound proteins, trypsin treatment should be employed.
  2. Clarity of Figure 3:
    In Figure 3, red fluorescence signals were observed in tissues treated with Atto550-labeled amiDNApds-1. Does this imply that Atto550-labeled amiDNApds-1 alone can enter plant cells without CPP? Higher-resolution images for panels 3a, 3b, and 3c should be provided to improve clarity and interpretation of results. In addition, agrobacteria harboring a GFP construct can be infiltrated into the leaves two days before CPP/GSA application so that subcellular localization of Atto550-labeled amiDNApds-1 can be determined.
  3. Concentration of CPPs:
    The authors used relatively high concentrations of CPPs and fusion proteins, approximately 100 μM for the five CPPs mentioned in Table 2. What is the minimum effective concentration of CPPs required for successful delivery of GSAs? Determining this would provide valuable insights into the practical application and cost-effectiveness of this technology.
  4. Phenotypic Changes in Treated Plants:
    Although the study demonstrated effective gene silencing, the lack of consistent results in the second experiment (Figure 6) raises concerns. Despite PDS gene knockdown, no visible physiological changes, such as photobleaching, were observed. This may limit the applicability of the method for field conditions, where visible phenotypic changes are critical for assessing the effectiveness of treatment.
  5. Repeated Foliar Applications:
    The study applied GSAs via repeated foliar treatments over four consecutive days (Figure 9), testing CPP alone, CPP + ASO Pool C (ASO-7 to ASO-12), CPP + amiRNA Pool (amiRNAPDS-1 to amiRNAPDS-5), or alternating between CPP + ASO Pool C and CPP + amiRNA Pool. The alternating application resulted in significant PDS gene knockdown compared to the CPP-only control. However, the requirement for repeated treatments increases costs, potentially limiting practical and economical field applications. Addressing ways to optimize the application schedule or reduce frequency would strengthen the study's applicability.
  6. Challenges with Monocot Gene Targeting:
    The presence of PDS paralogs and splice variants in Phragmites australis could complicate effective gene silencing, as current GSAs may not comprehensively target all variants necessary for phenotypic changes. A discussion of these challenges and potential solutions, such as designing more specific or broad-spectrum GSAs, would enhance the manuscript.

Author Response

Comment 1. Control in Figure 2:
The lack of an appropriate control in Figure 2 is a concern. A negative control, such as FAM-labeled dsRNA without CPP, should be included when counting the number of fluorescent dots. This would confirm whether surface proteins have been effectively removed. Additionally, the authors used a 10% phosphate buffer to wash the leaves, which removes only loosely bound surface proteins. For complete removal of surface-bound proteins, trypsin treatment should be employed.

Response 1: We totally understand the reviewer’s concern. In fact, we performed many more experiments than those presented in the manuscript using either FAM- or Atto550-labelled GSAs with or without FICT-labelled or non-labelled CPP. In most of these experiments, we included a GSA only (without CPP) group as one of the control treatments (see the newly added panels (l) to (o) in Figure 3 for example). None of these experiments showed detectable internalization of labelled GSAs if not conjugated with a CPP as the delivery vehicle. So, in the study presented in Figure 2, we did not include the FAM-labelled dsRNA without CPP as the control. Instead, we used the IM (infiltration medium) only as the negative control (background). We observed in this study that the internalization of dsRNA:CPP was dependent on treatment time, i.e., the longer the exposure, the more signals (fluorescent dots) we detected. Such characteristics fit well with the time/energy-dependent and rate-limiting endocytosis, the main route for CPP-mediated cellular internalization.

We appreciate the reviewer pointing out that trypsin, a proteolytic enzyme, can detach surface-bound proteins by cleaving them at the C-terminal side of lysine and arginine. However, we were not concerned about the residual leaf surface-bound CPP and CPP:GSA conjugates because we only counted for those internalized CPP or CPP:GSA in the leaf tissue cross-section slides.

Comment 2. Clarity of Figure 3:
In Figure 3, red fluorescence signals were observed in tissues treated with Atto550-labeled amiDNApds-1. Does this imply that Atto550-labeled amiDNApds-1 alone can enter plant cells without CPP? Higher-resolution images for panels 3a, 3b, and 3c should be provided to improve clarity and interpretation of results. In addition, agrobacteria harboring a GFP construct can be infiltrated into the leaves two days before CPP/GSA application so that subcellular localization of Atto550-labeled amiDNApds-1 can be determined.

Response 2: First, we replaced the original images in panels (h) to (k) with much better images showing much brighter red fluorescence signals in the red field (j) and the bright/green/red merged field (k). These images were taken for Atto550-labelled amiDNAPDS-1 complexed with non-labelled γ-zein-CADY at a much higher GSA:CPP ratio (1:6.25), than that (1:50) in the original images (h to k) as well as in panels (d) to (g). The increase of GSA concentration relative to CPP concentration led to more intensified signals (see also Figure 4). To demonstrate that the Atto550-labelled amiDNAPDS-1 (GSA) can’t enter plant cells without a CPP as the vehicle, we added four more panels (l to o) with images displaying no fluorescence signal for the GSA only treatment.

Panels 3a, 3b and 3c are indeed high-resolution images, clearly showing signals in three fields, green, red, and green/red merged. The green field showed a large amount of green dots representing FITC-labelled CPP, the red field showed many red dots representing Atto550-labeled GSA, and the merged field showed yellow dots representing the CPP:GSA complexes.

We are unsure how the use of agrobacteria can help with visualizing the subcellular location of GSA cargo on CPP. It is unknow whether any of the existing Agrobacterium strains can infect Phragmites australis. We do know that the agroinfiltration technique can be used to delivery plasmid constructs into plant cell nuclei. The plasmid can be engineered to harbor both GFP and GSA inserts. Even if we can successfully deliver a GFP- & GSA-co-expressing construct via agroinfiltration and determine the subcellular co-localization of expressed GFP and GSA transcripts, we don’t think that this would provide any evidence for the subcellular localization of GSA (amiRNA or dsRNA) delivered by the CPP platform. This is because the GFP- & GSA-co-expressing construct is delivered into nuclei for transcription, whereas the CPP directly delivers the RNA transcripts (GSAs like amiRNA and dsRNA) into cytoplasm to suppress the target PDS gene. Perhaps the reviewer could provide more clues on the suggested approach using an agroinfiltration-based platform.

Comment 3. Concentration of CPPs:
The authors used relatively high concentrations of CPPs and fusion proteins, approximately 100 μM for the five CPPs mentioned in Table 2. What is the minimum effective concentration of CPPs required for successful delivery of GSAs? Determining this would provide valuable insights into the practical application and cost-effectiveness of this technology.

Response 3: The application rate of 5 nmol CPP/leaf was determined in preliminary experiments, leading to a significant and sufficient amount of internalized CPPs and their payloads. We implemented this application rate throughout the present study. We added this information in lines 315-316 on page 17. Furthermore, it is worth noting that the delivery of GSAs is dependent on two factors: (1) the net charges of GSAs and CPPs, and (2) loading capacity set by the CPP:GSA ratio of the complexation (see the newly added text in lines 335 to 339 on page 18). We demonstrated that GSAs could be delivered effectively when the amount of CPP applied per leaf was 5.5 nmol and the CPP:GSA charge ratio falls between 12.5:1 and 2:1, (see Figures 3-5). We did not alter the amount of CPP applied per leaf in our experiments. Instead, we varied the amount of GSA to determine the appropriate range of charge ratios. If one needs to increase the amount of internalized GSA, it can be practically achieved by adjusting the CPP:GSA ratio and increasing the total amount of CPP:GSA complex.

Comment 4. Phenotypic Changes in Treated Plants:
Although the study demonstrated effective gene silencing, the lack of consistent results in the second experiment (Figure 6) raises concerns. Despite PDS gene knockdown, no visible physiological changes, such as photobleaching, were observed. This may limit the applicability of the method for field conditions, where visible phenotypic changes are critical for assessing the effectiveness of treatment.

Response 4: We are fully aware of and understand the concern raised by the reviewer. In fact, we have clearly stated the limitations of our findings and constrains we faced when this work was performed. It is obvious that the work presented in the manuscript represents our early foundational efforts and much more work needs to be done. In the Discussion and Conclusions sections, we discussed the future perspectives and directions for developing a field deployable, target-specific and environmentally benign RNAi-based gene silencing technology for genetic biocontrol of Phragmites and other invasive plant species.

Comment 5. Repeated Foliar Applications:
The study applied GSAs via repeated foliar treatments over four consecutive days (Figure 9), testing CPP alone, CPP + ASO Pool C (ASO-7 to ASO-12), CPP + amiRNA Pool (amiRNAPDS-1 to amiRNAPDS-5), or alternating between CPP + ASO Pool C and CPP + amiRNA Pool. The alternating application resulted in significant PDS gene knockdown compared to the CPP-only control. However, the requirement for repeated treatments increases costs, potentially limiting practical and economical field applications. Addressing ways to optimize the application schedule or reduce frequency would strengthen the study's applicability.

Response 5: As alluded in Response 4 above, the CPP:GSA platform developed in the present study is far from being applicable to the field. There remains a lot of work to be done. Even if it is matured, the RNAi-based gene silencing technology may still require repeated treatment within the same growing season or annually. We intentionally restrained ourselves from speculating future application scenarios for this embryonic technology.

Comment 6. Challenges with Monocot Gene Targeting:
The presence of PDS paralogs and splice variants in Phragmites australis could complicate effective gene silencing, as current GSAs may not comprehensively target all variants necessary for phenotypic changes. A discussion of these challenges and potential solutions, such as designing more specific or broad-spectrum GSAs, would enhance the manuscript.

Response 6: We have accepted and added this point to the end of the first paragraph on page 32 in the Discussion section (lines 609-612). We also want to bring to the reviewer’s attention that we mentioned that the lack of expected phenotypic response may be attributed to the existence of two paralogs or alternative splice variants (page 30 lines 563-577) and discussed many other Phragmites-specific and general challenges for gene silencing-based biocontrol in the Discussion section.

Reviewer 3 Report

Comments and Suggestions for Authors

The manuscript by Ji et al. provides experimental evidence that cell-penetrating peptides (CPPs) can penetrate leaves of a monocot weed, and when the CPPs were complexed with compositions of gene silencing agents (GSAs), a small but statistically significant degree of PDS gene silencing could be detected. The study is noteworthy in that it demonstrates the potential of CPPs to administer herbicidal GSAs to control invasive plants/weeds. The authors note that many of the combinations of CPP-GSA treatments yielded inconsistent results, which they attributed to different technical issues. Nevertheless, in one experiment, they observed a significant and consistent (~20%) transcript knockdown.

The potential implications of the study are highly intriguing, but given the uncertainties in their findings, I think that more experiments need to be conducted to provide convincing evidence that the technology is robust enough for others to pursue this line of research.

Below I outline my concerns and ask that the authors consider some further work to strengthen the data to a point that will encourage others to corroborate their findings in other systems.

1.      In Figure 1, can the authors explain whether the dots of fluorescence represent single or aggregated CPPs, and are they being sequestered by the plant cell into intracellular compartments (e.g. vacuoles)? If they are internalized into compartments, could that limit their ability to mediated transcript knockdown/inhibition of translation?

2.      Can the authors explain why, in Figure 2, the level of internalized fluorescence of the continues to increase from day 1 to day 4? Does this imply that the CPPs take days to penetrate the leaves, or does this reflect the slow aggregation of CPPs (perhaps into cellular compartments)?

3.      The resolution of the images in Figure 3 is not sufficient to evaluate whether there is co-localization of the different fluorescent tags, to convince readers that the CPP is still linked to the GSA. Please enlarge and label the images.

4.      In Figure 6a, where a significant reduction in PDS transcripts was detected at 5 DPT, it appears that the level of DPT is also significantly higher in the negative control, relative other days. This increase thus makes it appear that a knockdown has occurred, but is this just a sampling issue? Can the authors account for this increase in PDS at day 5? This increase in PDS transcripts did not occur in the replicate experiment (Figure 6b), and hence, no knockdown was detected.

5.      The most compelling evidence for PDS transcript reduction occurred with a mixing of multiple different artificial microRNAs and multiple different antisense oligos, applied alternately over several days, which make it impossible to ascertain which GSA is the key effector. I appreciate that the authors were pleased to observe consistent down-regulation of the target transcript, but I think this experiment needs to be taken to completion to identify the optimal CPP-GSA combination.

Author Response

General comment. The potential implications of the study are highly intriguing, but given the uncertainties in their findings, I think that more experiments need to be conducted to provide convincing evidence that the technology is robust enough for others to pursue this line of research.

Response: We totally agree with the reviewer on this. As we repeatedly stated in the Discussion and Conclusions sections of the manuscript, the present study constitutes the foundational, proof-of-concept work that warrants and encourages much further work to enhance the GSA design, delivery, stability, scalability, cost-effectiveness, etc.

Comment 1. In Figure 1, can the authors explain whether the dots of fluorescence represent single or aggregated CPPs, and are they being sequestered by the plant cell into intracellular compartments (e.g. vacuoles)? If they are internalized into compartments, could that limit their ability to mediated transcript knockdown/inhibition of translation?

Response 1: Figure 1 shows the results of a rapid screening experiment where 5 in-house designed fusion or non-fusion CPPs were applied to Phragmites leaves ex vivo and the leaf cross-sections were prepared one-hour after CPP application. The fluorescence dots may represent aggregated CPP molecules, judged by their size and brightness. We can’t tell from these images (10× magnification) the exact cellular compartment location of the internalized CPPs. So, we modified the heading of the paragraph (page 16 line 311) from “Intracellular internalization of CPPs via topical application on P. australis leaf” to “Tissue uptake of CPPs via topical application on P. australis leaf” to align our interpretation better with the images in Figure 1. Furthermore, these images can’t answer the reviewer’s first question about CPP sequestration or trapping in any of the subcellular compartments such as vacuoles. As the reviewer may have noticed that we added an endosomal escape domain (EED) to the in-house designed fusion and non-fusion CPPs. It is widely accepted that as intracellular sorting organelles, endosomes in plant cells regulate trafficking of proteins and lipids among other subcellular compartments of the secretory and endocytic pathway, specifically the plasma membrane Golgi, trans-Golgi network (TGN), and vacuoles or lysosomes (see https://www.nature.com/scitable/topicpage/endosomes-in-plants-14404958/). The incorporation of an EED allows a CPP internalized through endocytosis to escape from being trapped in endosomes and transported into or sequestered within subcellular compartments. Instead, the CPP containing an EED (along with a GSA cargo) could be released into the cytoplasm (https://www.nature.com/articles/srep32301) so that the GSA could hybridize to and subsequently suppress the transcripts of its target gene. Therefore, we believe that the addition of EED to CPP designs can improve the intracellular delivery efficiency of the GSA payload.

The images in Figures 3-5 provide further evidence for the intracellular distribution of internalized CPP:GSA conjugates. So, we think that subcellular sequestration should not play a significant role in reducing the gene silencing effects of the GSAs tested in the present study.

Comment 2. Can the authors explain why, in Figure 2, the level of internalized fluorescence of the continues to increase from day 1 to day 4? Does this imply that the CPPs take days to penetrate the leaves, or does this reflect the slow aggregation of CPPs (perhaps into cellular compartments)?

Response 2: We don’t know exactly why it took 3-5 days for the applied CPP:GSA to penetrate the leaves (Figures 2-5). However, this observation corroborates with the modest (20-38%) but statistically significant suppression of target PDS gene expression (Figures 6a, 7a & 9). These results indicate that the internalization of CPP:GSA conjugates may be driven by rate-limiting, energy-dependent endocytosis, which may take days to process.

Comment 3. The resolution of the images in Figure 3 is not sufficient to evaluate whether there is co-localization of the different fluorescent tags, to convince readers that the CPP is still linked to the GSA. Please enlarge and label the images.

Response 3: We replaced the original images in panels (h) to (k) with much better images showing much brighter red fluorescence signals in the red field (j) and the bright/green/red merged field (k). These images were taken for Atto550-labelled amiDNAPDS-1 complexed with non-labelled γ-zein-CADY at a much higher GSA:CPP ratio (1:6.25), than that (1:50) in the original images (h to k) as well as in panels (d) to (g). The increase of GSA concentration relative to CPP concentration led to more intensified signals (see also Figure 4). Panels 3a, 3b and 3c are indeed high-resolution images, clearly showing signals in three fields, green, red, and green/red merged. The green field showed a large amount of green dots representing FITC-labelled CPP, the red field showed many red dots representing Atto550-labeled GSA, and the merged field showed yellow dots representing the CPP:GSA complexes. In addition, to demonstrate that the Atto550-labelled amiDNAPDS-1 (GSA) can’t enter plant cells without a CPP as the vehicle, we added four more panels (l to o) with images displaying no fluorescence signal in the images of the GSA only treatment.

Comment 4. In Figure 6a, where a significant reduction in PDS transcripts was detected at 5 DPT, it appears that the level of DPT is also significantly higher in the negative control, relative other days. This increase thus makes it appear that a knockdown has occurred, but is this just a sampling issue? Can the authors account for this increase in PDS at day 5? This increase in PDS transcripts did not occur in the replicate experiment (Figure 6b), and hence, no knockdown was detected.

Response 4: It seems that there are missing words in this comment. We attempt to respond to it based on our understanding, i.e., was the significant reduction in the relative PDS gene expression of the dsRNA treatment group at 5 DPT in Figure 6a attributed to a “random” increase in that of the control group? This result was not reproduced in another experiment (see Figure 6b). We attributed the difference between the two experiments to FAM-labelling that may protect dsRNA from being degraded by intracellular nucleases. This argument was based on the well-known instability of dsRNA and its vulnerability to enzymatic degradation. As for the variations of relative PDS/actin gene expression in the control group, we think they are within the reasonable range for each individual experiment (e.g., 2±0.5 in Figure 6, 8±2 in Figures 7-8, and 12±1 in Figure 9). Although significant variations between different experiments exist owing to seasonal environmental changes and plant development, both treated and control groups in each experiment used the same batch of plants, which would not affect statistical comparison between different treatment groups of the same experiment.

Comment 5. The most compelling evidence for PDS transcript reduction occurred with a mixing of multiple different artificial microRNAs and multiple different antisense oligos, applied alternately over several days, which make it impossible to ascertain which GSA is the key effector. I appreciate that the authors were pleased to observe consistent down-regulation of the target transcript, but I think this experiment needs to be taken to completion to identify the optimal CPP-GSA combination.

Response 5: Findings reported in this manuscript represent our early work in developing RNAi-based gene silencing technologies for invasive plant or weed control. As alluded in the manuscript, the biggest challenge we encountered was the incomplete target gene characterization (e.g., sequence, homologs, alternative splicing products, etc.). With improved genetic characterization of target genes, we have redesigned GSAs. Meanwhile, in addition to CPPs, we also explored other nanocarriers as the delivery vehicles. We have obtained much more consistent and more promising results which will be published in separate papers soon.

Round 2

Reviewer 2 Report

Comments and Suggestions for Authors

The authors have addressed my questions and the revised manuscript has been improved.

Author Response

We are happy to see that the reviewer is satisfied with our response to Round 1 comments and the revised manuscript.

Reviewer 3 Report

Comments and Suggestions for Authors

I thank the authors for their well-considered response to my questions, and with the revisions they have made, I think the manuscript is suitable for publication.

Author Response

We are very glad to learn that the reviewer thinks we adequately addressed the comments and that the revised manuscript is now acceptable for publication.